# A climatology of sub-seasonal temporal clustering of extreme precipitation in Switzerland and its links to extreme discharge

Alexandre Tuel[1] and Olivia Martius[1,2]

[1]Institute of Geography and Oeschger Centre for Climate Change Research, University of Bern, Switzerland
[2]Mobiliar Lab for Natural Risks, University of Bern, Switzerland

**Correspondence:** Alexandre Tuel (alexandre.tuel@giub.unibe.ch)

**Abstract.** The successive occurrence of extreme precipitation events on sub-seasonal timescales can lead to large precipitation accumulations and extreme river discharge. In this study, we analyse the sub-seasonal clustering of precipitation extremes in Switzerland and its link to the occurrence and duration of extreme river discharge. We take a statistical approach based on Ripley's K function to characterise the significance of the clustering for each season separately. Temporal clustering of precipitation extremes exhibits a distinct spatio-temporal pattern. It occurs primarily on the northern side of the Alps in winter and on their southern side in fall. Cluster periods notably account for 10-16% of seasonal precipitation in these two regions. The occurrence of a cluster of precipitation extremes generally increases the likelihood and duration of high-discharge events compared to non-clustered precipitation extremes, particularly at low elevations. It is less true in winter, when the magnitude of precipitation extremes is generally lower, and much of the precipitation falls as snow. In fall, however, temporal clusters associated with large precipitation accumulations over the southern Alps are found to be almost systematically followed by extreme discharge.

## 1 Introduction

Switzerland's climate, topography and high population density make floods one of the major natural disasters, accounting for instance for 71% of weather-related insurance claims over the 1973-2011 period (Swiss Re, 2012) and 36% of total damages to buildings between 1995 and 2014 (BAFU, 2016). Apart from high Alpine regions where summer snowmelt accounts for a large part of the flood hazard, floods and landslides over much of Switzerland typically follow widespread heavy precipitation (Froidevaux et al., 2015; Froidevaux and Martius, 2016). Large precipitation accumulations may sometimes result from the occurrence of several extreme precipitation events in close succession. In contrast to persistent but moderate wet conditions, temporal clusters of extreme precipitation events involve more than one day of extreme precipitation. Such events can lead to extreme river discharge, flash flooding (Doswell et al., 1996) or mass movement, especially in urban and mountain areas (Guzzetti et al., 2007; Panziera et al., 2016). Temporal clustering of extremes also complicates rescue, clean-up and repair efforts (Raymond et al., 2020). Furthermore, clusters of extremes tend to be missing from risk models which often rely on assumptions of independence in the timing of extreme precipitation occurrence (Priestley et al., 2018).

In Switzerland, several major floods in recent history were linked to series of extreme precipitation events (Barton et al.,

2016). In 1993, three events that occurred between September 21 and October 15 led to record 10- to 30-day precipitation accumulations in southern Valais and the Ticino region, and caused repeated overflowing of Lake Maggiore above its 100-year return level. Similar conditions around Lake Maggiore were repeated over the course of four weeks in fall 2000, and again in November 2002, each time bringing the lake above its critical flooding level. The major August 2005 floods in central Switzerland were likewise connected to a series of heavy rainfall events in the second half of the month (BAFU and WSL,

2008). Recent other examples of temporal clusters of precipitation extremes leading to major floods include the Pakistan floods of summer 2010 (Martius et al., 2013), the Central Europe floods of summer 2013 (Grams et al., 2014) and the UK floods of winter 2013/2014 (Priestley et al., 2017).

Assessing the temporal dependence in the occurrence of extreme precipitation events at the catchment scale is therefore crucial to accurately quantify flood hazard. Such an assessment has not yet been attempted for the whole of Switzerland.

Barton et al. (2016) analysed the sub-seasonal serial/temporal clustering of precipitation extremes in Southern Switzerland using a non-parametric approach, Ripley's K function, and found a significant tendency toward clustering during the fall season. They also examined the weather dynamics associated with specific cluster events. At the European scale, Yang and Villarini (2019) quantified the influence of large-scale climate modes on the temporal clustering of extreme precipitation, while Mailier et al. (2006); Vitolo et al. (2009) and Pinto et al. (2013) looked at clustering in winter extratropical storms in the

Euro-Atlantic sector, a region where serial cyclone clustering is particularly relevant (Dacre and Pinto, 2020). More recently, Tuel and Martius (2021) attempted a systematic global and seasonal assessment of extreme precipitation temporal clustering at sub-seasonal timescales. Their analysis was however conducted at coarse spatial resolutions relative to the size of Swiss catchments, and focused primarily on clustering signals at large spatial scales.

Additionally, few studies have attempted to evaluate the links between extreme discharge or floods and extreme precipitation

clusters, and none has conducted such a systematic assessment for Switzerland. The analysis by Barton et al. (2016) focused on selected examples of floods triggered by clusters of extreme precipitation events, as did other studies over Europe (Blackburn et al., 2008; Grams et al., 2014; Huntingford et al., 2014; van Oldenborgh et al., 2015; Priestley et al., 2017; Insua-Costa et al., 2019) or Southwest Asia (Martius et al., 2013). Villarini et al. (2013) considered the temporal clustering of flood events over the American Midwest and its link to large-scale climate patterns, but did not discuss precipitation. Kopp et al. (2021), by contrast,

presented a global perspective on the link between sub-seasonal clustering and extremes of cumulative precipitation. In their analysis, however, Switzerland was covered by 3-4 major catchments only, which prevented a discussion of local variability in the results.

Relationships between extreme precipitation and flood occurrence in Switzerland have been more extensively analysed. Stucki et al. (2012) and Froidevaux and Martius (2016) both looked at atmospheric precursors of extreme floods, and Giannakaki and

Martius (2016) discussed the weather patterns associated with intense precipitation events. Helbling et al. (2006) and Diezig and Weingartner (2007) assessed the role of different flood drivers across Swiss catchments, like extreme or continuous rainfall, rain on snow events, and snow and/or glacier melt. Finally, Froidevaux et al. (2015) quantified the influence of accumulated precipitation before annual peak discharge events across 101 Swiss catchments. They showed that short-range antecedent precipitation, up to three days before an event, was the most relevant predictor of peak discharge occurrence and magnitude.

Long-range antecedent precipitation, from 4 days to a month before an event, was nevertheless still relevant for the Jura mountains and parts of the Swiss Plateau. However, it is unclear whether these conclusions still hold in the case of prolonged or recurrent high-discharge conditions, characterised by repeated exceedances of daily discharge percentiles over short time windows. The hydrological response to multiple extreme precipitation events occurring as part of a cluster may also differ from the response to the same events separated by longer time periods. Paschalis et al. (2014) indeed showed that discharge peaks were strongly shaped by antecedent soil wetness conditions, themselves largely affected by the temporal correlation of precipitation. In addition, in basins with high retention capacities (*e.g.*, with natural or artificial lakes), single extreme precipitation events may not be enough to trigger extreme discharge, unlike prolonged periods of heavy precipitation.

The goals of this study are therefore twofold. First, we aim to quantify the sub-seasonal clustering of precipitation extremes in time across Switzerland using several gridded station- and satellite-based datasets. We take the same approach as Barton et al. (2016) and Tuel and Martius (2021) which relies on Ripley's K function as an indicator of clustering, and analyse each season separately to remove the seasonal signal in extreme precipitation magnitude. We then discuss the patterns and robustness of the spatio-temporal distribution of sub-seasonal clustering in Switzerland. Second, we evaluate the links between extreme precipitation clusters, extreme precipitation accumulations and extreme discharge using observed discharge data at 93 gauges across Switzerland. After introducing the data and methods used in this study, we briefly discuss the seasonality of extreme precipitation and discharge magnitude across Switzerland, before moving on to results and their discussion.

## 2   Data and study area

### 2.1   Data

#### 2.1.1   Precipitation

Reference precipitation data for this study comes from the daily 2x2km RhiresD dataset. RhiresD, developed by MeteoSwiss, covers the period from 1961 to present. It is obtained by spatial interpolation of data from a high-density rain-gauge network that extends across Switzerland, with at least 420 stations available for any single day. The effective scale of RhiresD varies as a function of station density, but is on average of the order of 15-20 km, the typical inter-station distance. A detailed description of this dataset can be found at https://www.meteoswiss.admin.ch/home/climate/swiss-climate-in-detail/raeumliche-klimaanalysen.html and the interpolation algorithm is described in Frei and Schär (1998).

For purposes of comparison, we also consider other daily precipitation datasets, at their native resolutions: ERA5 (Hersbach et al., 2020), the latest ECMWF reanalysis available from 1979 onwards at 0.25° resolution, in which precipitation is a forecasted quantity, *i.e.* not directly constrained by assimilated observations; the satellite-based TRMM TMPA (TRMM Multi-Satellite Precipitation Analysis) 3B42 version 7 (50°S-50°N, 1998-2019, 0.25° resolution) (Huffman et al., 2007), CMORPH (60°S-60°N, 2003-2019, 0.25° resolution) (Joyce et al., 2004); and the land-only, station-based Climate Prediction Center Global Unified Gauge-Based Analysis of Daily Precipitation (1979-2019, 0.5° resolution) (Chen et al., 2008) and EOBS gridded product version 19.0e (1950-2019, 0.25° resolution) (Haylock et al., 2008).

### 2.1.2 Discharge observations

We analyse daily discharge observations for 93 small to medium-sized gauged catchments (14-1700 km$^2$) distributed across Switzerland (Figure 1-a). The catchments cover a wide variety of catchment characteristics and climates, from glacial and nival runoff regimes at high altitudes to pluvial regimes in the Swiss plateau (see Table A1 for catchment characteristics). These catchments were analysed by Muelchi et al. (2021b) who selected them based on several criteria: data availability, the absence of major lakes, minimal human influence and satisfactory calibration results in their hydrological model. The data for each catchment range from January 1961 to December 2017 (Table A1); the proportion of catchments with data rises from about 45% in the early 1960s to more than 95% in 1995, at which level it remains until 2015, before rapidly decreasing to 30% by the end of 2017. For each catchment, the analysis is conducted over the period for which discharge data is available. This means that daily discharge percentiles (and precipitation percentiles, when precipitation and discharge are considered together) are calculated over different time periods depending on the catchment.

### 2.1.3 Catchment-scale aggregation

We average RhiresD data over a hydrological partitioning of Switzerland that consists of 63 catchments with a mean area of 900 km$^2$ (see Figure 4). Catchment-scale aggregation is useful to identify the occurrence of high-impact heavy precipitation events, and also to smooth RhiresD data to a lower resolution more consistent with its effective resolution. Though we could also use the set of 93 gauged catchments, this set does not cover the whole of Switzerland. Consequently, we opt for a countrywide partitioning of 63 larger catchments (for which no discharge observations are available). To be comprehensive and to help compare results between discharge and precipitation, we also show in the appendix the results obtained for the 93-catchment set (Figures A4, A5 and A6).

### 2.2 Study area

Switzerland can be divided into several regions with distinct climates and hydrological regimes: the Jura, the Plateau, the Alps and the Southern Alps (Figure 1-b) (Umbricht A, 2013; Aschwanden and Weingartner, 1985). These regions notably exhibit quite different seasonal cycles in extreme precipitation and discharge occurrence. In the Plateau, the heaviest precipitation occurs chiefly during summer (Figure 2-c) (Helbling et al., 2006; Diezig and Weingartner, 2007; Panziera et al., 2018), as a result of convective instability (Stucki et al., 2012), frequent westerly winds and Atlantic water vapour transport (Giannakaki and Martius, 2016). In summer, however, evapotranspiration is highest and soils are less saturated than in the cold season. Consequently, extreme discharge events are about equally likely to occur in winter, spring and summer (Figure 3). In the Jura, while the magnitude of extreme precipitation events still peaks in summer, its seasonality is less pronounced. About 20% of extreme precipitation events indeed occur in winter and spring each (Figure 2), triggered by forced orographic ascent of moist westerlies (Froidevaux and Martius, 2016). Extreme discharge, however, is mostly confined to winter and spring, largely driven by rain-on-snow processes (Diezig and Weingartner, 2007; Helbling et al., 2006; Köplin et al., 2014).

As in the Jura, the seasonal cycle in extreme precipitation occurrence over the Alps is not strong (Figure 2) (Frei and Schär,

1998; Umbricht A, 2013). The peak is reached in summer and fall for most catchments, when extreme precipitation occurs as a result of local convective instability (Stucki et al., 2012), but winter and spring still concentrate 30-40% of extreme events. The outlook for discharge is very different, however. Alpine catchments, especially at high elevations, are mainly driven by snow- and glacier melt (Aschwanden and Weingartner, 1985). Thus, extreme discharge is almost exclusively confined to summer (Figure 3-c) (Köplin et al., 2014; Muelchi et al., 2021a). Finally, the Southern Alps experience extreme precipitation mostly during summer and fall (Figure 2-c,d) (Frei and Schär, 1998; Isotta et al., 2014). Such behaviour results from the frequent southerly advection of moist Mediterranean air caused by upper-level troughs (Barton et al., 2016). These atmospheric conditions are connected to potential vorticity streamers or cut-offs centred west of the Alps, which are most frequent during fall (Martius et al., 2006). Extreme discharge in this region also occurs primarily during fall (50-60% of events; Figure 3-d).

## 3 Methods

A summary of the methodology adopted in this study is shown on Figure A1.

### 3.1 Precipitation and discharge extremes

For each dataset, precipitation extremes are defined on a monthly basis as days when daily accumulated precipitation exceeds the 99th percentile of the corresponding month. For instance, January precipitation values are compared to the January 99th percentile. The percentiles are calculated using all days (both with and without precipitation). Potential trends in extreme daily precipitation percentiles are not taken into account. As the individual weather systems associated with extreme precipitation may sometimes last for several days, we remove the short-term temporal dependence in the occurrence of extreme precipitation events by applying a standard runs declustering procedure (Coles, 2001) with a run length of 2 days, well-suited for Switzerland Barton et al. (2016). The goal of the declustering is to remove short-term dependence and to identify independent events. This procedure is for example applied prior to a peak-over-threshold statistical analysis. The declustering merges extreme events that are separated by less than 2 days into a single event. Consequently, it reduces the number of extreme events compared to the original series. Events within each season (winter: DJF; spring: MAM; summer: JJA; and fall: SON) are then analysed together.

Extreme discharge events for each of the 93 gauged catchments are defined as days when daily discharge exceeds its 95th or 99th percentile calculated on the whole time series (both percentiles are analysed separately). We look at two extreme percentiles to test the sensitivity of our results to the choice of threshold, and also to increase the number of detected high-discharge events. The persistence in extreme discharge conditions is assessed by identifying periods of persistent high discharge over sub-seasonal timescales. These are defined as periods of length $L$ containing at least $N$ extreme discharge days. Three sets of $(L, N)$ values are considered: $\{(10, 5), (20, 8), (30, 10)\}$. Depending on the values of $L$ and $N$, no periods may be found in some catchments, in which case they are simply excluded from the corresponding analysis.

Three reasons justify the choice of seasonally-varying thresholds for precipitation and fixed thresholds for discharge. First, such a choice removes the influence of the seasonality in extreme precipitation magnitude. The occurrence rate of extreme

precipitation events is therefore constant across the year, and detecting clustering significance is straightforward (see below). Second, impacts of discharge extremes are usually related to their absolute rather than relative magnitude. Third, the seasonal cycles of extreme precipitation and discharge magnitudes are not in phase over much of Switzerland (Figures 2 and 3). The most extreme discharge does not necessarily occur after the heaviest precipitation events. Surface conditions, like soil satu-

160 ration, presence of snow/ice, vegetation cover, or evaporative demand, considerably shape the discharge response to heavy precipitation (Paschalis et al., 2014). As they vary substantially from one season to the next, the discharge response to the same precipitation magnitude may differ depending on the season.

## 3.2  Sub-seasonal temporal clustering of precipitation

Temporal clustering of precipitation extremes is quantified with Ripley's K function (Ripley, 1981). We give here a quick
overview of the methodology and refer the reader to Tuel and Martius (2021) for further details. For a given window size $w$, Ripley's K function applied to a time series measures the average number of extreme events in a neighbourhood of $w$ days before and after a random extreme event in the series. This gives information about the tendency towards temporal clustering in the series. The larger the value of Ripley's K function for a given $w$, the more clustered the extreme events. The significance of temporal clustering in the series is then assessed by comparing Ripley's K values to those obtained from a Monte-Carlo sample
of 5000 simulated homogeneous Poisson processes with the same average event density as the observed series. In homogeneous Poisson processes, events occur independently from each other and therefore exhibit complete temporal randomness. Because we chose monthly percentiles to define extreme precipitation events, the occurrence rate of extremes is constant throughout the year. Thus we can test for clustering significance against homogeneous series. Non-homogeneous (and more complex) series would have been required if the likelihood of extreme event occurrence has been a function of time.

From this comparison we get an empirical p-value for each $w$. As we deal with multiple hypothesis tests, we implement a false discovery rate procedure (Wilks, 2016) with a baseline significance level of 5% to identify catchments where clustering is significant. Clustering significance is assessed for two intervals of $w$ values, characteristic of sub-seasonal timescales: 15-25 and 25-35 days. Clustering is said to be significant for a given interval if it is significant for at least half of the $w$ values in that interval.

## 3.3  Identification of cluster events

We identify extreme precipitation cluster events over 21-day time windows with the algorithm of Kopp et al. (2021) (see their Figures 3 and 4). Starting from the declustered binary extreme event series, the first step is to calculate the 21-day moving sum of extreme event counts. In a second step, we select the 21-day period with the largest event count (*i.e.*, the highest number of extreme events), if that count is larger than 2. Otherwise, no clusters are found and the algorithm stops. In the case of multiple
21-day periods with the same extreme event count, the one with the largest precipitation total is selected first. In the third step, we remove from the binary event series the extreme events that occur in the selected 21-day period. The algorithm is then run again from the first step onwards to identify the next cluster event. This procedure avoids any overlap between cluster events. The choice of the 21-day time window is well-suited to quantify clustering at sub-seasonal timescales, and is generally

consistent with the length of observed cluster episodes that led to major floods in Switzerland (see introduction). Results do
not differ significantly for slightly shorter or longer (2-4 weeks) windows (see also Kopp et al. (2021)).

We then characterize clusters of precipitation extremes with two metrics related to their potential impact. The first is the average
contribution of cluster periods to seasonal precipitation. This contribution increases with the frequency and total precipitation
of cluster periods. The second metric is the frequency of cluster periods during extreme 21-day precipitation accumulations. It
gives an idea of how often cluster periods are responsible for extreme precipitation accumulations, a frequent trigger of flood
events in Switzerland (Froidevaux et al., 2015).

### 3.4 Effects of temporal clustering of extreme precipitation on the occurrence and duration of extreme discharge

We analyse the influence of clusters of precipitation extremes on discharge in two ways: by looking at discharge characteristics
after clusters of precipitation extremes, and at precipitation characteristics before persistent high-discharge periods.

First, we calculate for each catchment the probability $p_1$ of extreme discharge for all days during and up to 5 days after 21-day
clusters of precipitation extremes. We also calculate the probability $p_2$ of extreme discharge days for periods of the same length
and same time of the year as the selected cluster periods. We can then define an odds ratio of extreme discharge occurrence
after clusters of extremes as $\frac{p_1(1-p_2)}{p_2(1-p_1)}$ (Wilks, 2019). The odds ratio compares the likelihood of extreme discharge occurrence in
the presence of a precipitation cluster to its likelihood in the absence of a precipitation cluster. The higher it is, the stronger the
relationship between the occurrence of extreme discharge and precipitation clusters. From the identification of 21-day cluster
periods, precipitation extremes can also be separated into "clustered" and "non-clustered" events. We then look at the likelihood
of extreme discharge occurrence after both types of events to highlight potential differences in the discharge response.

Second, for each of the persistent high-discharge periods identified as described previously, we calculate the number of pre-
cipitation extremes and the percentile of total accumulated precipitation over a window stretching from 10 days before the
beginning of the persistent high-discharge period to its end. We choose to begin 10 days before because Froidevaux et al.
(2015) showed that moderate wet conditions occurred in the week preceding many flood events in Switzerland, hence the need
to look beyond the few days preceding persistent high-discharge periods.

## 4 Results

### 4.1 Spatio-temporal patterns of clustering significance

We now turn to the analysis of Ripley's K values and their implications in terms of sub-seasonal temporal clustering. Several
coherent areas exhibit $K$ values that are significantly larger than those expected for homogeneous Poisson processes with no
temporal dependence (Figure 4). In winter, significant temporal clustering of precipitation extremes is mainly found in central
Switzerland, along the Alpine ridge, at the 15-25 and 25-35 day timescales (Figures 4-a; A2-b). In spring, two catchments in
Northern Switzerland as well as a few catchments in Southeastern Switzerland also exhibit significant clustering (Figures 4-b;
A2-d). By contrast, results for the summer season show a complete absence of temporal clustering significance at all timescales

(Figures 4-c; A2-e,f). Finally, in fall, significance is found at all timescales over both the western tip of Switzerland and the southern side of the Alps (Figures 4-d; A2-g,h).

Similar patterns are found by comparing to the coarser-resolution precipitation datasets (ERA5, TRMM, EOBS, CPC and CMORPH), with some notable exceptions. Clustering significance over the Alps in winter is also present in the coarser-resolution data, but with a wider extent than in RhiresD (Figure 5-a, see also Figure A3). Temporal clustering during spring is

generally less significant across Switzerland (Figure 5-b). Two datasets indicate significant clustering locally in northwestern Switzerland, but none do along the northern and southern borders where significance was found in RhiresD. By contrast, all datasets agree on the absence of clustering in summer (Figure 5-c), and on significant clustering in southern and southeastern Switzerland during the fall season (Figure 5-d). Temporal clustering does not appear particularly significant, however, in western Switzerland during fall. In winter, clustering significance extends over a large region stretching from the Mont Blanc

massif in France to eastern Switzerland along the Alpine ridge, in good agreement with RhiresD (Figures 4-a and 5-a). Similarly, southern Switzerland is part of a larger region exhibiting significant clustering, encompassing northern Lombardy and possibly extending southwards to the Mediterranean shore (Figure 5-d).

### 4.2   Characteristics of cluster events

We now expand the statistical analysis by showing the characteristics of clustered precipitation events. In winter, extreme

precipitation clusters contribute an average of $\approx$10% to total winter precipitation along the Alpine ridge where clustering is statistically significant (Figure 6-a). Additionally, clusters occur during about 60-70% of extreme 21-day precipitation accumulations (above the corresponding $99^{\text{th}}$ percentile) (Figure 7-a). Elsewhere, clusters contribute little both to seasonal and extreme precipitation accumulations. In spring, the average contribution of clusters to seasonal precipitation is overall weak ($<$10%), even for catchments where clustering in RhiresD is statistically significant (Figure 6-b). Yet, over Western Switzer-

land, periods of extreme 21-day accumulations are almost always cluster periods as well (Figure 7-b). In summer, consistent with the absence of clustering at that time of the year, clusters are not contributing much to seasonal precipitation. Finally, in fall, cluster contribution to seasonal precipitation reaches its annual maxima of 12-16% over Southeastern Switzerland (particularly the Southern Alps). It is also quite high ($\geq$10%) over Western Switzerland where clustering is statistically significant as well (Figure 6-d). In addition, more than 80% of extreme precipitation accumulation periods are accompanied by cluster events

in the Southern Alps (Figure 7-d). Since extreme discharge in this area are most common during fall (Figure 3-d), this suggests a possibly important role of extreme precipitation clusters in high-impact weather events in this region and at that time of the year.

### 4.3   Discharge response to extreme precipitation clustering

Persistent high-discharge periods, regardless of $L$ and $N$ values, are systematically associated with extreme precipitation

accumulations for catchments with mean elevations up to about 1500m (Figures 8-a,b and 9-a). Above 1500m, glacial/nival runoff regimes dominate and the link to extreme precipitation accumulations is weaker. By contrast, extreme precipitation clusters precede persistent high-discharge periods only in the Southern Alps and locally over the eastern parts of the Swiss

Plateau and the Jura (Figure 8-c,d). The dependence to catchment elevation is similar, with a much weaker intersection of cluster events and extreme discharge at high elevations, but less robust with a larger spread of values at low elevations (Figure 9-b).

For most catchments, particularly those below 1500m elevation, the discharge response after an extreme precipitation event differs between single extreme events and events that are part of clusters (Figure 10). In the five days following an extreme precipitation event, the fraction of days exceeding either the 95[th] or the 99[th] percentiles of daily discharge is higher when that event belongs to a cluster. The difference is particularly large over northern Switzerland and in the Southern Alps where, for instance, the 99[th] percentile of daily discharge values is exceeded on average 20-30% of the time (1-1.5 days) in the five days following an extreme precipitation event during a cluster, but only 10-15% if the extreme occurred outside of a cluster. Given that daily discharge exceeds its 99[th] percentile on average only $\approx$3.5 days a year, this implies an important effect of clustered extremes. The occurrence of a cluster of precipitation extremes greatly increases the likelihood of high-discharge events, particularly at low elevations, as evidenced high odds ratio values (Figure 11). Daily discharge is more than 7 (respectively 10) times more likely to exceed its 95[th] (respectively 99[th]) percentile during and after a cluster in the Jura, for instance. The distribution of daily discharge percentiles after extreme precipitation events confirms these results: while the probability of exceeding high-discharge thresholds on the day following an extreme event is not very different between clustered and non-clustered extremes, that probability decreases much faster in the days following non-clustered extremes (Figure 12).

## 5  Discussion

### 5.1  Patterns of clustering significance and their physical interpretation

Our definition of extreme precipitation events based on the exceedance of monthly percentiles of daily precipitation removes the influence of seasonality in extreme precipitation magnitude (Figure 2). By applying Ripley's K function to the resulting time series, we can thus focus on short-term temporal dependence in extreme precipitation occurrence driven by sub-seasonal dynamics or intra-annual variability (*e.g.*, climate modes). Clustering significance is found mainly over the Alps during winter and in southern and southeastern Switzerland during fall (Figures 4 and A2). The spatial coherence and robustness of the results over these two regions across timescales and datasets (Figure 5) suggests that specific physical processes are responsible for the clustering.

Both in winter or in fall, clustering significance is chiefly concentrated at timescales below 30 days, which seems to preclude any dominant role of seasonal sea-surface temperature anomalies (Tuel and Martius, 2021).

First, we discuss the spatial pattern during winter, when significant clustering occurs in the central Alps. During winter, extreme precipitation events in northern Switzerland often occur in connection with extreme integrated water vapour transport (e.g., linked to atmospheric rivers) that interacts with the orography (Piaget et al., 2015; Froidevaux and Martius, 2016; Giannakaki and Martius, 2016). The concentration of clustering significance over the Alps during winter possibly results from a spatial anchoring of precipitation at high elevations, through orographic lifting and convergence of the moist airmasses, while precipitation in the lower-lying area is spatially more dependent on the presence of cold-air pools upstream of the mountains

(Medina and Houze, 2003; Rössler et al., 2014; Piaget, 2015). The spatial extent of these cold pools may vary from event to event, along with the location of the strongest precipitation.

Next, we discuss the spatial pattern during fall when clustering occurs south of the Alps. During fall, several major clusters of precipitation extremes south of the Alps were related to recurrent Rossby wave breaking over western and southwestern Europe (Barton et al., 2016). The wave breaking leads to enhanced low-level moisture transport from the Mediterranean towards the Alps (Martius et al., 2006; Barton et al., 2016). Barton et al. (2016) discuss why wave breaking was recurrent for four case studies. Recurrent cyclogenesis and extratropical transition of tropical cyclones upstream over the western Atlantic seem to play a role, but not necessarily a systematic one. Persistence in blocking conditions in the northwestern Atlantic also contributed to upper-level wave amplification and the subsequent occurrence of precipitation extreme clusters in Southern Switzerland (Barton et al., 2016).

The apparent lack of agreement among precipitation datasets regarding the significance of the clustering over western Switzerland in fall (Figures 4-d and 5-d) is less straightforward to interpret. Despite the precautions taken to control the false discovery rate, it is still possible that significance in this region is detected by pure chance. Yet, it is not altogether obvious that the gridded satellite or reanalysis datasets are completely reliable either. If thunderstorms are responsible for many extreme precipitation events their representation in reanalysis products can be questioned. Their small spatial scale may also cause them to be missed in satellite products. While coarse-resolution gridded datasets seem to agree on the timing and magnitude of precipitation extremes in western Switzerland during fall (Rivoire et al., 2021), a more detailed comparison with RhiresD data and an analysis of the type of events responsible for these extremes would be required to conclude.

## 5.2 Relevance of extreme precipitation clusters for flood hazard

Persistent high-discharge periods at elevations lower than 1500m a.s.l. are almost systematically associated with extreme precipitation accumulations in the preceding days (Figure 8-a,b). This is consistent with the mainly pluvial regimes at lower elevations, as well as with the results of Froidevaux et al. (2015). Such accumulations are not always the consequence of clusters of extremes, however. Outside southern Switzerland during fall, western Switzerland during spring and, to a lesser extent, Alpine catchments during winter, the overlap of cluster events and extreme precipitation accumulations is generally smaller than 50%, and often smaller than 25% (Figure 7). Results in summer and spring can be understood by the scarceness of cluster events in those seasons. In winter, despite being frequent, clusters over the Alps are less systematically associated with extreme accumulations, which are often the result of a single heavy precipitation event.

Still, from the perspective of surface impacts, clusters remain relevant, regardless of their overall frequency, if they increase flood hazard. The discharge response to both clustered and non-clustered extreme precipitation events typically peaks one day after the event (Figure 12), consistent with the findings of Froidevaux et al. (2015). However, our results show that clusters of precipitation extremes increase the likelihood of occurrence and the duration of high-discharge events, particularly at low elevations (Figures 10 and 11). This influence is noticeably larger than for non-clustered precipitation extremes (Figure 12). On average, daily accumulated precipitation during clustered and non-clustered extremes is similar. Instantaneous precipitation rates might be different, but it is not possible to verify it given the daily resolution of the precipitation data. However, the first

extreme in a cluster event likely increases soil moisture, which enhances the discharge response to the subsequent precipitation extremes (Merz et al., 2006; Nied et al., 2014; Paschalis et al., 2014). The role of antecedent soil moisture on flood generation and volume is well-documented for Switzerland and Alpine catchments (e.g., Keller et al., 2018). This may explain why extreme discharge probability decreases more slowly after clustered precipitation extremes compared to non-clustered events (Figure 12).

This difference is quite high in the Southern Alps (*e.g.*, Figure 9-c,d), possibly due to the fact that floods in this area generally occur in the fall (Figure 3-d; Barton et al. (2016)) when clusters bring substantial amounts of precipitation (Figure 7-d). There, frequent clusters leading to extreme precipitation accumulations are likely to be an important precursor of major flood events, as confirmed by observations of several damaging clustering periods (Barton et al., 2016). This region of Switzerland also experiences the largest precipitation extremes (Umbricht A, 2013). Additionally, it is characterised by poor infiltration rates, steep slopes and weak soils (Aschwanden and Weingartner, 1985). Infiltration excess (connected to Hortonian-type storm runoff generation) may therefore be more rapidly reached than in the rest of the country. Coupled with saturation excesses following the first extreme event in a cluster, it might explain why the region stands out in most of our analyses.

By contrast, in the Alps during winter, though clustering is statistically significant, its impact on extreme discharge is quite limited. This results most likely from the fact that discharge in Alpine catchments is lowest in winter, when much of the precipitation falls as snow and the magnitude of precipitation extremes is generally lower.

Finally, the case of Western Switzerland during spring is interesting. Though rare, clusters are responsible for almost all extreme precipitation accumulations (Figure 7-b). Over this region, floods are somewhat less frequent in spring than in winter (Figure 3-a,b), despite similar extreme precipitation likelihood (Figure 2-a,b). This may result from fewer rain-on-snow events, a major flood process for the region (Aschwanden and Weingartner, 1985; Köplin et al., 2014) but also drier soils coupled to high infiltration rates (Aschwanden and Weingartner, 1985). Yet, spring floods can still be quite devastating, since precipitation generally falls as rain instead of snow, and limited vegetation cover makes erosion more likely. Consequently, cluster events that affect Western Switzerland during spring should be the focus of further research.

### 5.3 Some limitations and future prospects

Our approach to quantify links between clustered extremes and flood response has a number of limitations. First, we based our analysis on the exceedance of given precipitation and discharge percentiles, not taking into account either flood volumes or precipitation intensities. This is naturally quite restrictive, and in particular fails to capture potential non-linearities in relationships above the selected percentiles. In addition, spatial variability in extreme discharge or average total cluster precipitation were not analysed.

Our results only focus on the hazard component of flood risk. We do not take into account exposure and vulnerability, which may differ substantially between catchments due to variability in population density, infrastructure, flood management capacities, etc. Similarly, we did not take into account the influence of catchment regulation in this work. While the analysed catchments are generally not heavily regulated ones, human influence may still be felt, especially when it results in the smoothing over time of extreme discharge conditions, which would impact our analysis of persistent flood events.

Finally, our definition of floods and flooding persistence was also somewhat simplistic. First, while from the perspective of
impacts it makes sense to define floods based on annual discharge percentiles, in snow-driven or glaciated catchments, this
choice may discard potential high-discharge conditions occurring outside summer. Second, our definition of flooding persistence based on a minimal number of flood days in a given time window lumps together single, long floods and recurrent short
ones, two kinds of events with potentially different impacts, and which may have to be managed differently.

## 6 Conclusions

The main findings of this study are as follows. First, we identified a specific seasonal and spatial pattern of significance
in sub-seasonal temporal clustering of extreme precipitation events across Switzerland. Various station- and satellite-based
datasets point to generally significant clustering over the Alps in winter, particularly their central part, and over Southern
Switzerland during fall. Second, extreme precipitation clusters play a contrasted role in seasonal and sub-seasonal extreme
precipitation accumulations. Their contribution is particularly high in fall over southern Switzerland, but more limited over
the Alps in winter. Clusters are also frequently associated with extreme precipitation accumulations over Western Switzerland
during spring, despite their relative scarcity. Finally, cluster events, regardless of their frequency, are generally associated with
a higher flood likelihood and more persistent flood conditions over much of Switzerland. The Southern Alps region stands out
from this analysis. There, clusters of precipitation extremes are frequent during fall and tend to bring particularly large amounts
of precipitation. As a result, they appear to be critical precursors of major flood events, a conclusion supported by previous
event-based analyses. While our results are exclusively focused on Switzerland, the method adopted for this analysis could in
principle be applied to other regions of the world to quantify the relevance of temporal heavy precipitation clusters for flood
hazard.

*Data availability.* The RhiresD dataset is provided by MeteoSwiss, the Swiss Federal Office of Meteorology and Climatology. Discharge
data were obtained from Switzerland's Federal Office for the Environment. ERA5 reanalysis data for the 1979-2019 period are available from https://apps.ecmwf.int/datasets/. TRMM data can be downloaded at https://disc.gsfc.nasa.gov/datasets/TRMM_3B42_Daily_7/
summary. CMORPH data are provided by NOAA's National Centers for Environmental Information at https://www.ncei.noaa.gov/data/
cmorph-high-resolution-global-precipitation-estimates/access/daily/0.25deg/. CPC Global Unified Precipitation data is provided by the NOAA/OAR/ESR
PSL, Boulder, Colorado, USA, from their website at https://psl.noaa.gov/.

*Author contributions.* O.M. designed and supervised the research. A.T. designed the research, implemented the code, analysed the data and
provided the figures; A.T. and O.M. wrote the manuscript.

*Competing interests.* The authors declare no competing interests.

*Acknowledgements.* The authors gratefully acknowledge the Swiss Federal Office of the Environment (FOEN) and Regula Mülchi for the Swiss river discharge data. O.M. acknowledges support from the Swiss Science Foundation (SNSF) grant number 178751.

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

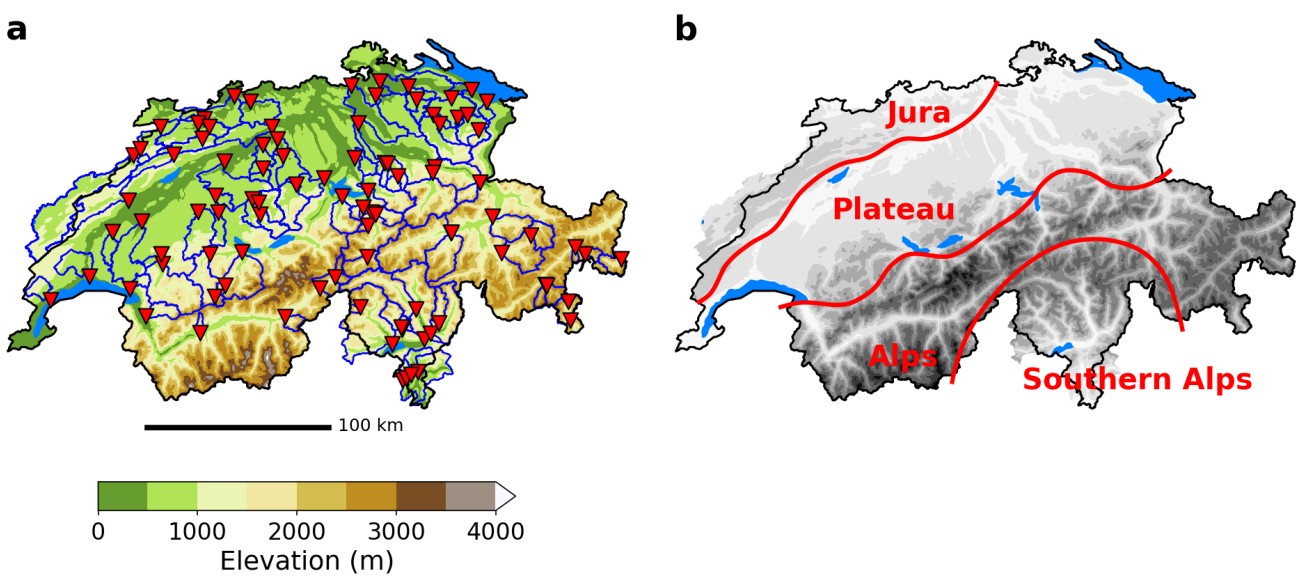

**Figure 1.** (a) Topography of Switzerland (shading, with major lakes shown in light blue) and gauged catchments used in this study (catchment boundary: blue lines; catchment gauge location: red triangles). The thick black line indicates the Swiss border. (b) Switzerland's topography (shaded) and major climate/hydrological regions (red).

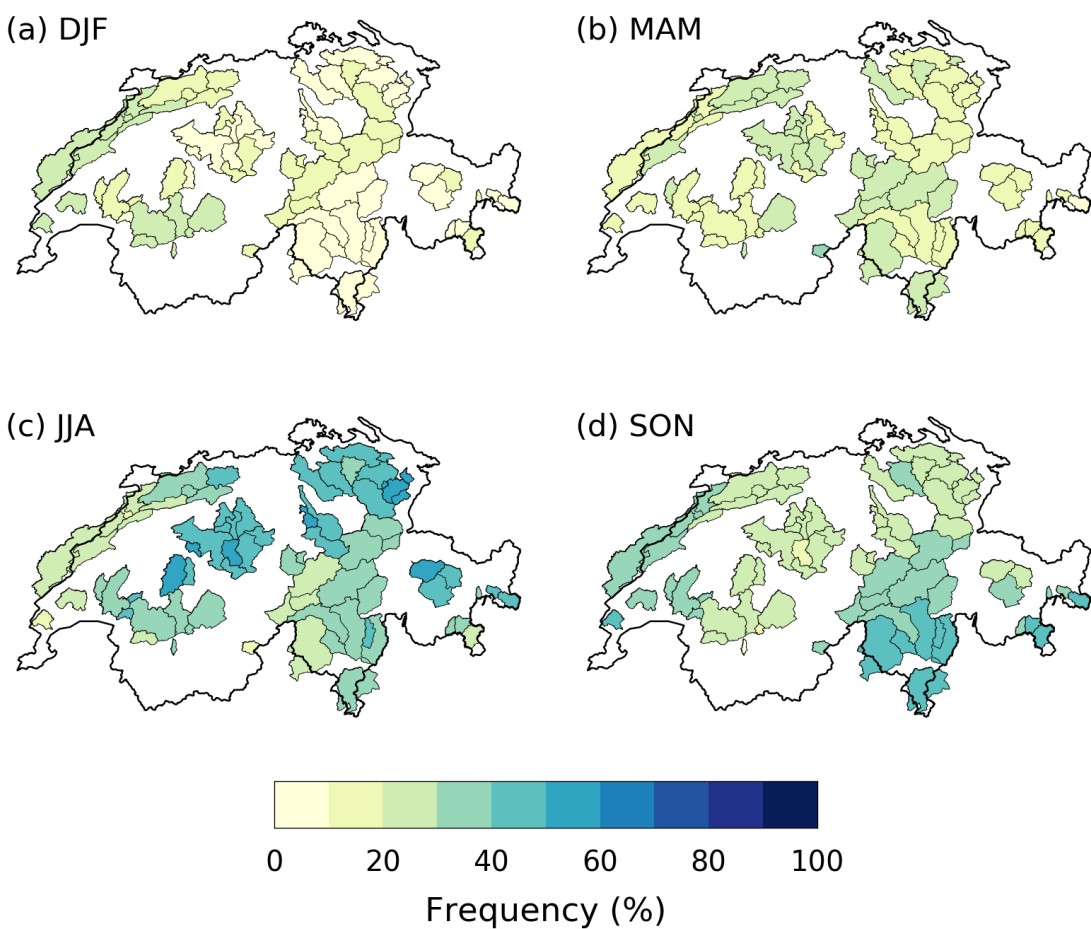

**Figure 2.** Seasonal frequency of exceedance of annual 99[th] daily precipitation percentile in RhiresD: (a) DJF, (b) MAM, (c) JJA and (d) SON.

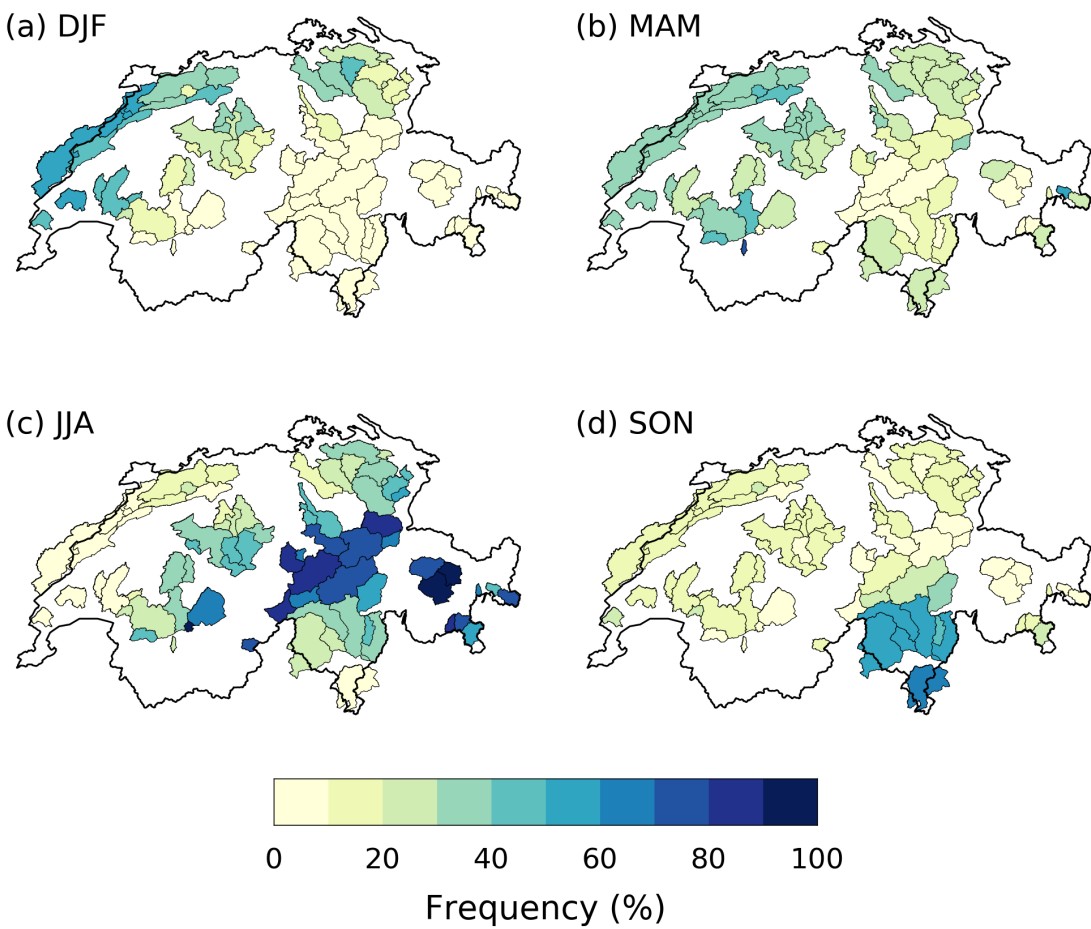

**Figure 3.** Seasonal frequency of exceedance of annual 99$^{th}$ daily discharge percentile: (a) DJF, (b) MAM, (c) JJA and (d) SON.

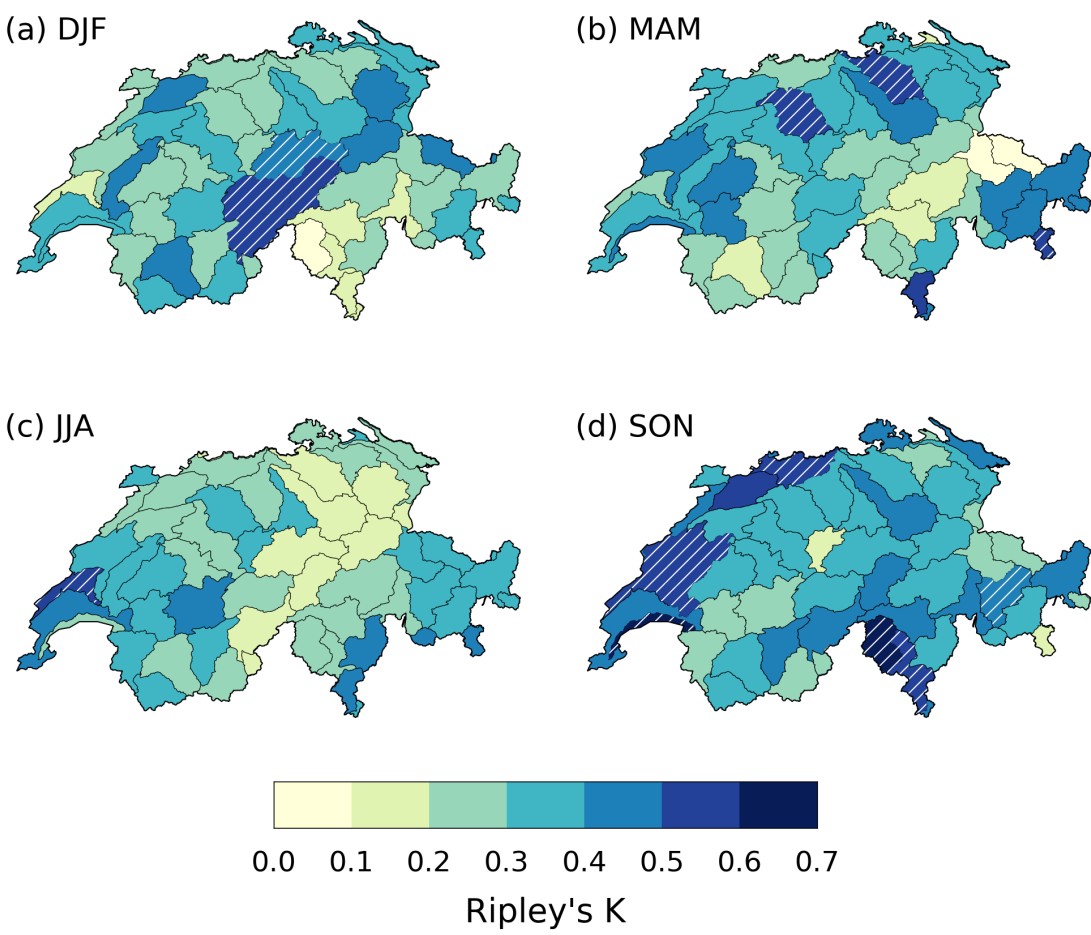

**Figure 4.** Value of Ripley's K in the RhiresD dataset for a 20-day window (shaded), and clustering significance for a time window of 15-25 days (white hatching) in (a) DJF, (b) MAM, (c) JJA and (d) SON.

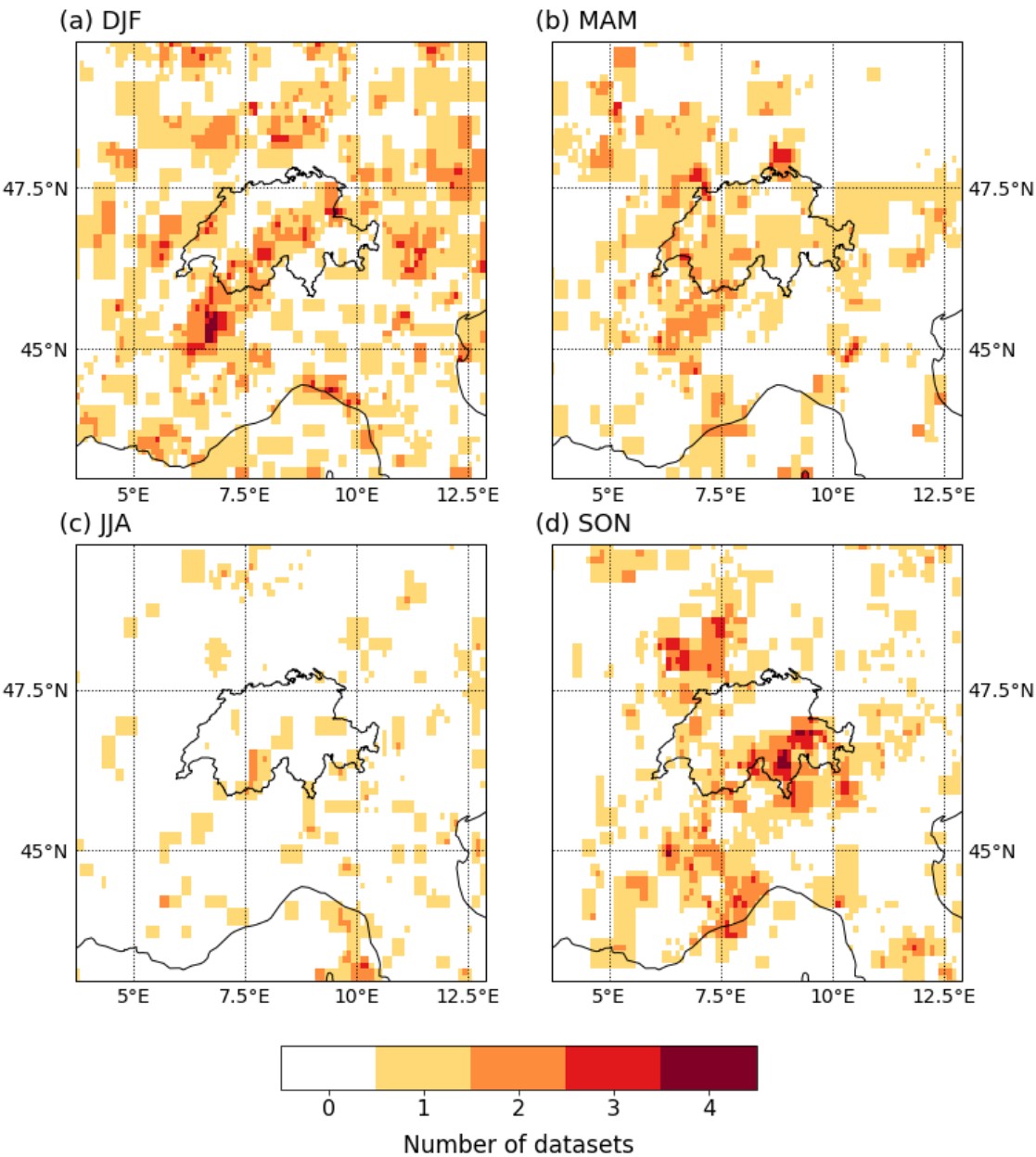

**Figure 5.** Number of datasets (among ERA5, TRMM, CMORPH, CPC and EOBS) that agree on the significance of extreme precipitation clustering for a 15-25 day window in (a) DJF, (b) MAM, (c) JJA and (d) SON. For this comparison, all datasets were regridded to the smallest 0.1° EOBS resolution.

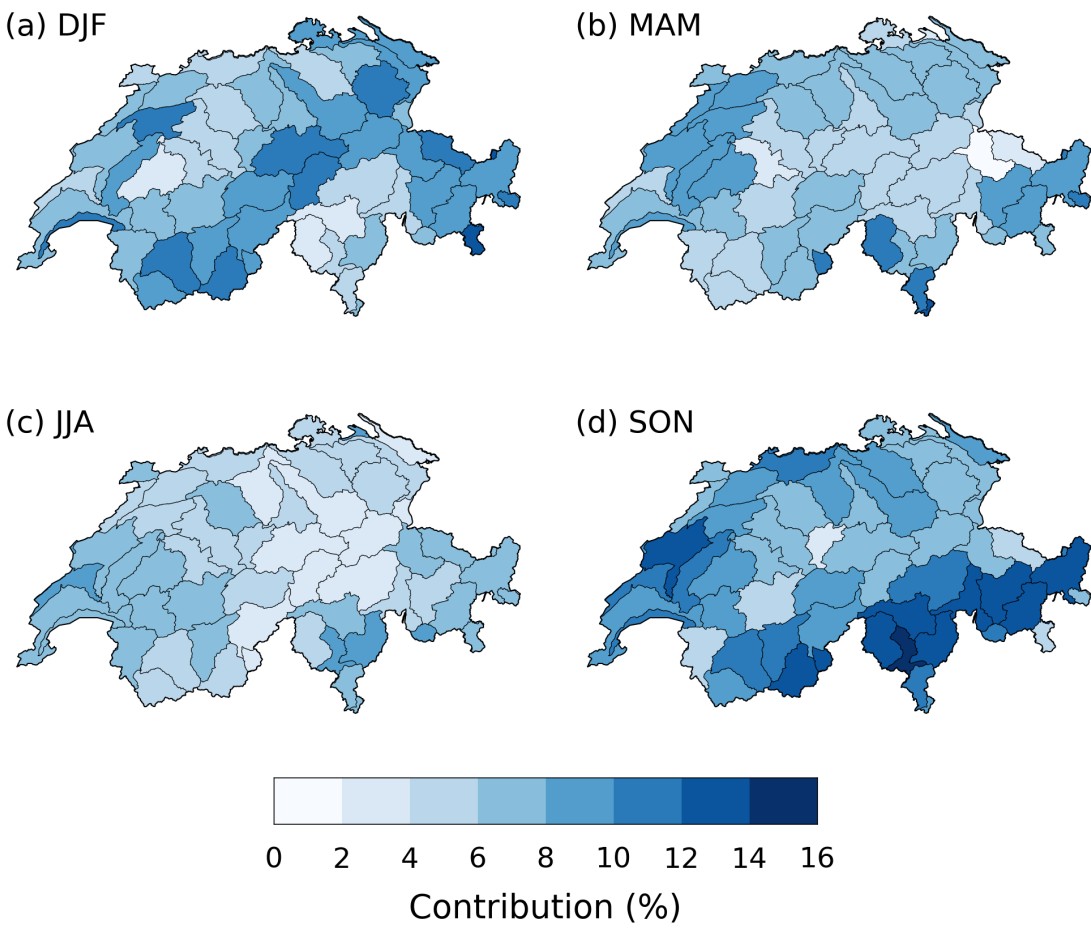

**Figure 6.** Contribution of 21-day extreme precipitation clusters to seasonal precipitation in RhiresD, in (a) DJF, (b) MAM, (c) JJA and (d) SON.

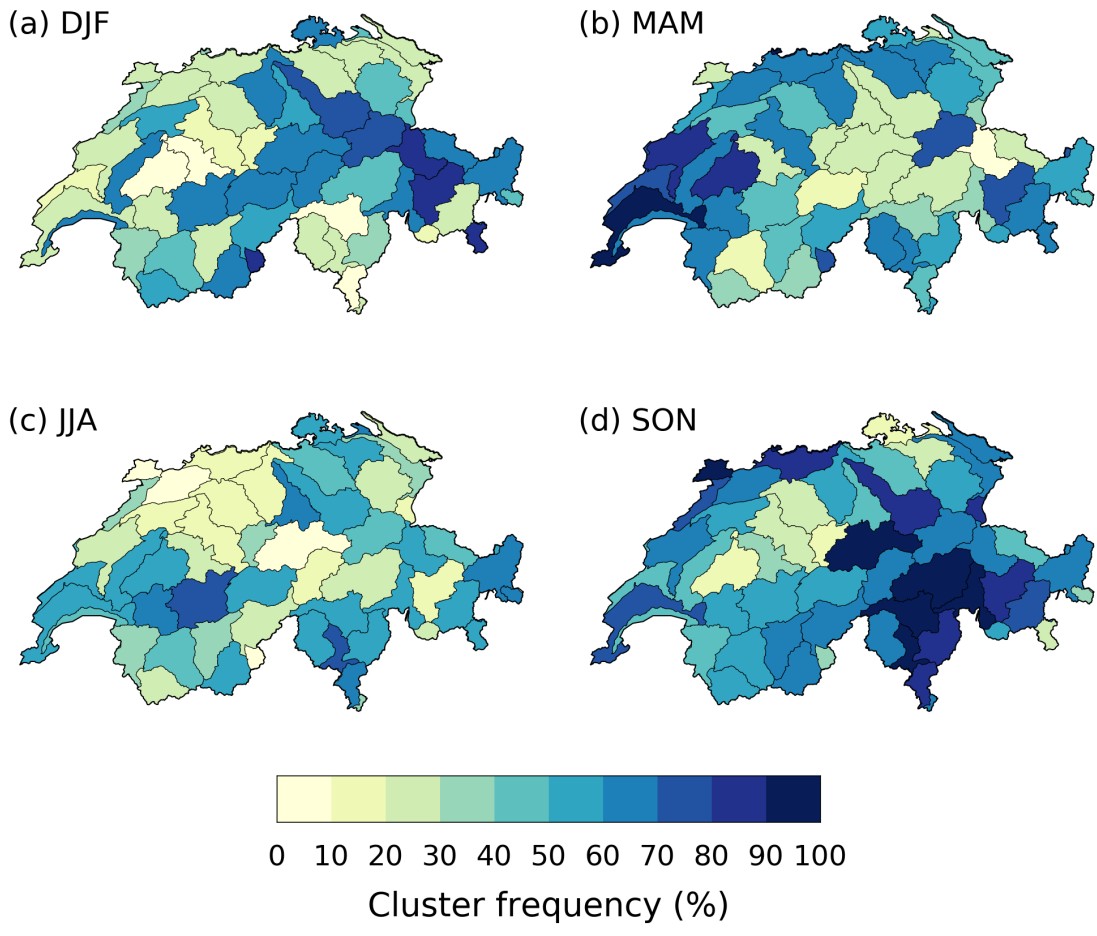

**Figure 7.** Frequency of extreme precipitation cluster occurrence during extreme 21-day cumulative precipitation events ($> 99^{\text{th}}$ percentile) in RhiresD, in (a) DJF, (b) MAM, (c) JJA and (d) SON.

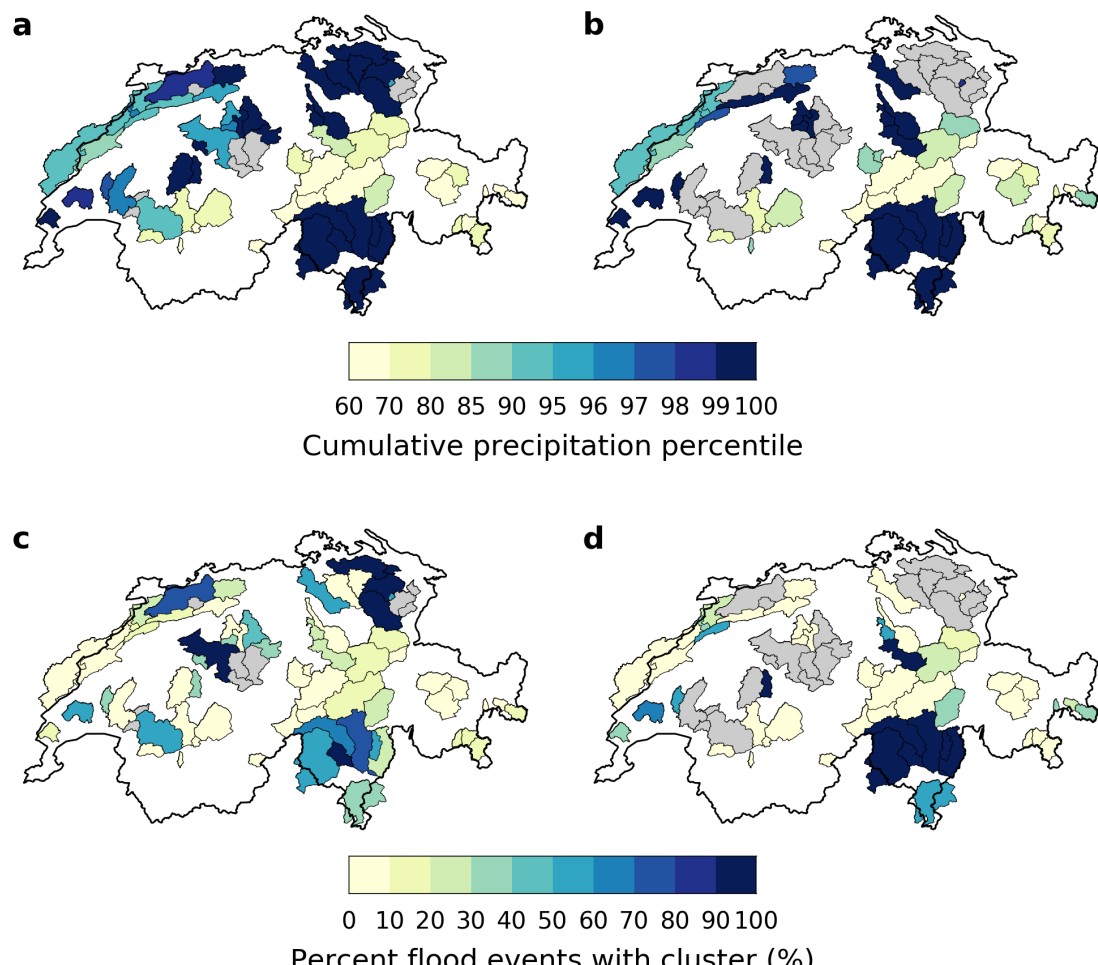

**Figure 8.** (a,b) Average cumulative precipitation percentile during and 10 days prior to persistent high-discharge periods, defined as: (a) at least 5 days within a 10-day window with discharge above its 99th percentile, and (b) at least 8 days within a 20-day window with discharge above its 99th percentile. Catchments for which such persistent high-discharge periods are not observed are shown in white. (c,d) Same as (a,b), but for the frequency of cluster occurrence during and 10 days prior to persistent high-discharge periods.

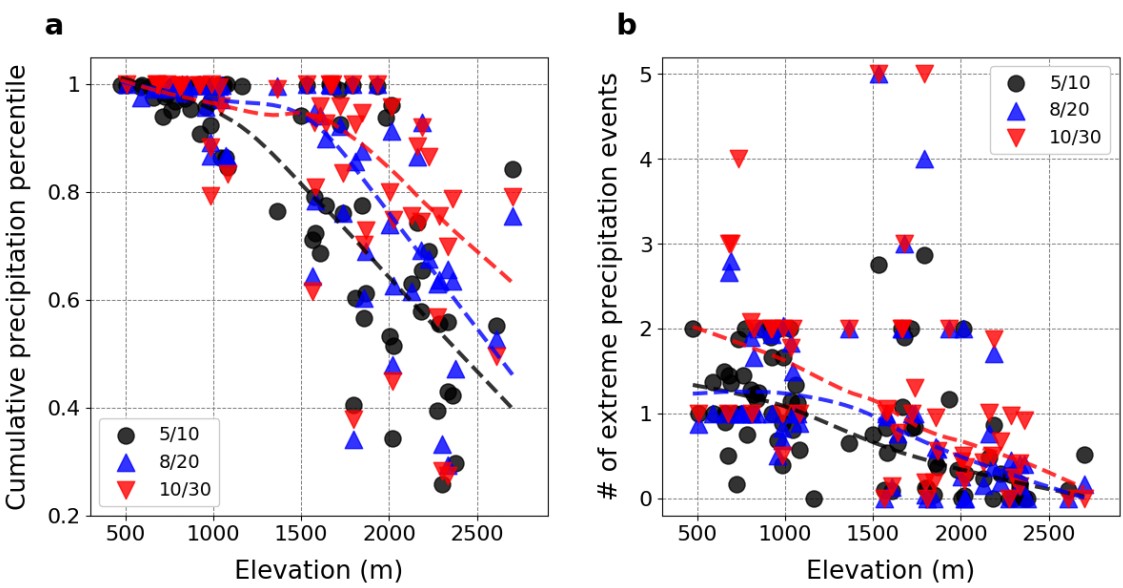

**Figure 9.** (a) Average cumulative precipitation percentile during and 10 days prior to persistent high-discharge periods as a function of mean catchment elevation. Black circles (respectively blue triangles, red triangles) correspond to events characterized by at least 5 (respectively 8, 10) days within a 10-day (respectively 20-day, 30-day) window with discharge above its $99^{\text{th}}$ percentile. (b) Same as (a), but for the number of extreme precipitation events.

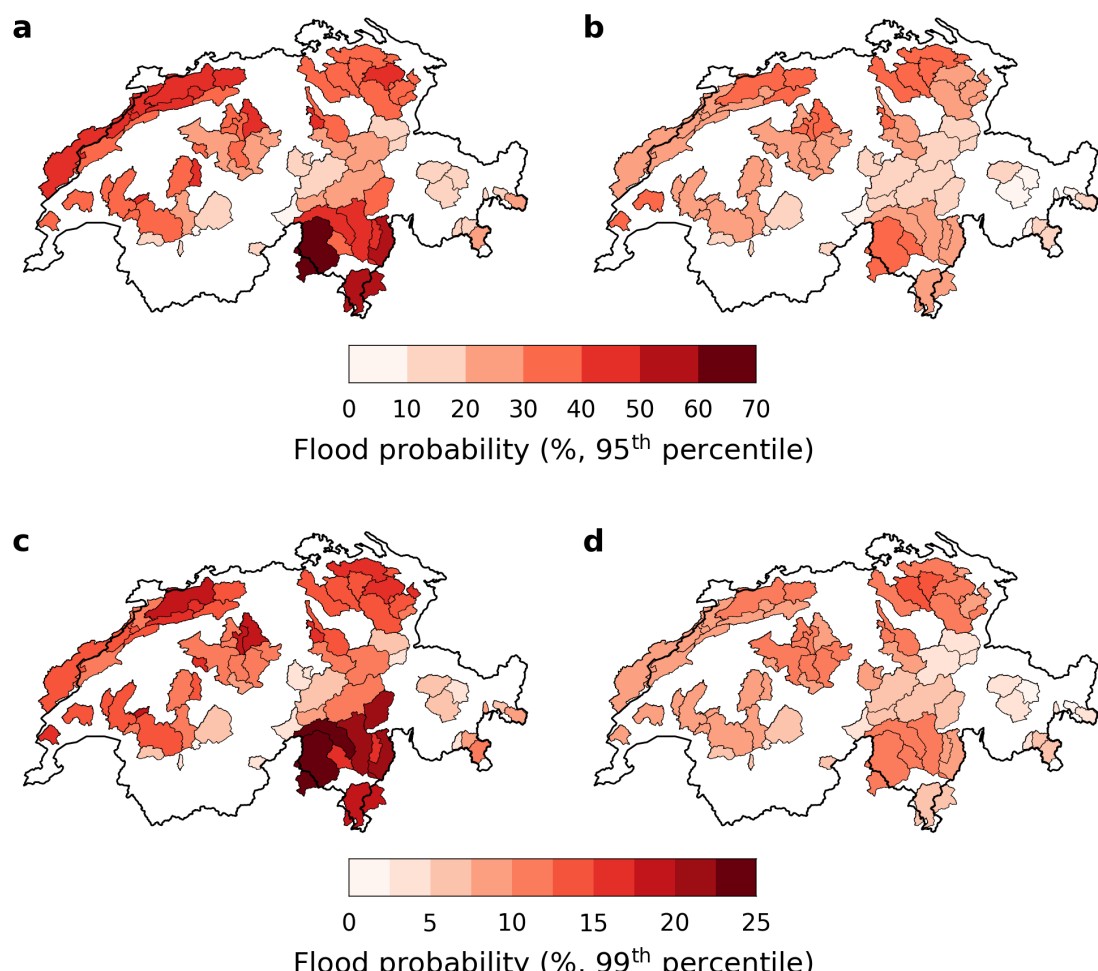

**Figure 10.** (a,b) Fraction of days with discharge above its $95^{th}$ percentile in the 5 days following (a) an extreme precipitation event that is part of a 21-day cluster and (b) any random precipitation extreme. (c,d) Same as (a,b) but for the $99^{th}$ daily discharge percentile.

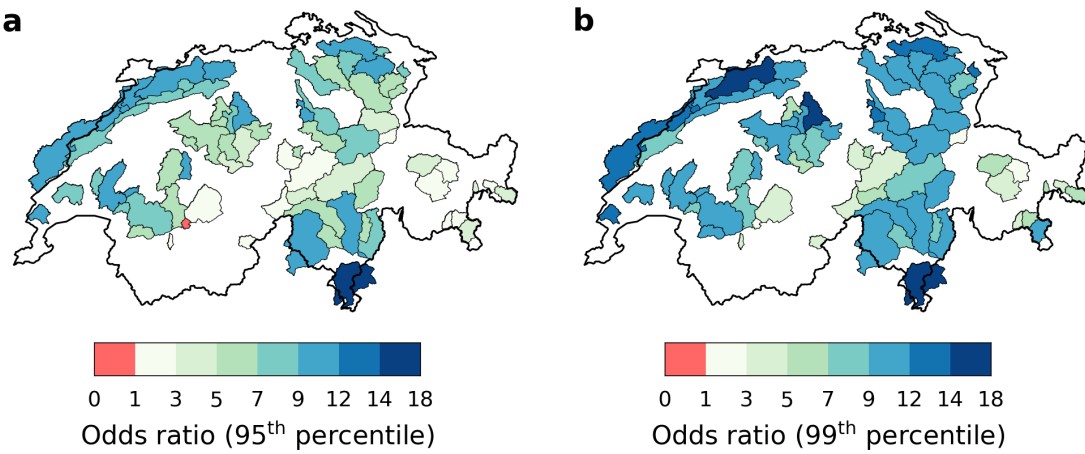

**Figure 11.** Odds ratio for extreme discharge occurrence (defined based on daily (a) 95[th] and (b) 99[th] discharge percentiles) during and up to 5 days after cluster events (see section 3.4 for methodological details).

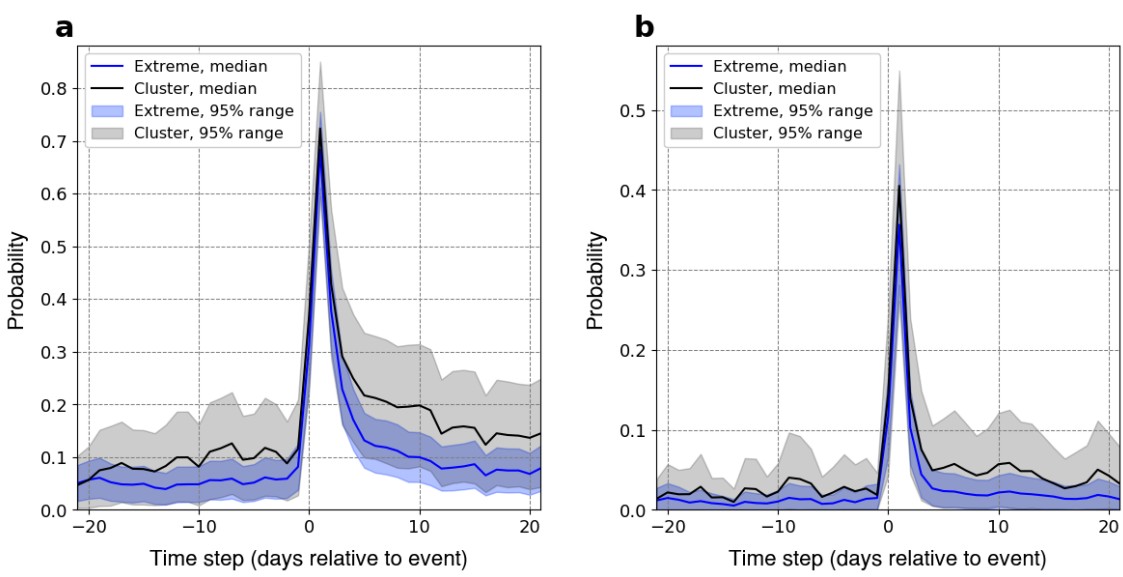

**Figure 12.** Probability of exceeding the (a) $95^{\text{th}}$ and (b) $99^{\text{th}}$ percentile of daily discharge around an extreme precipitation event that is part of a 21-day cluster (black) and around any random precipitation extreme (blue) across catchments with a mean elevation smaller than 1500m. Solid lines correspond to the multi-catchment median and shading to the 95% range.

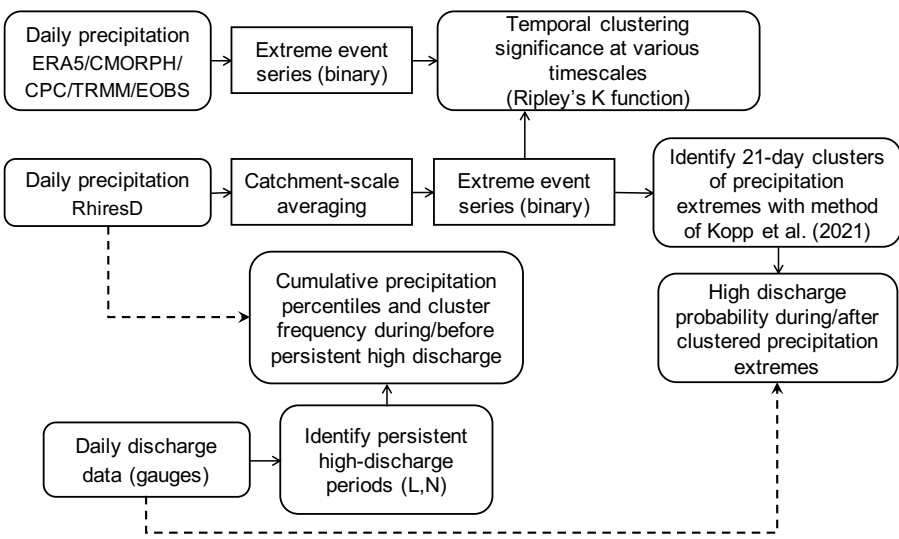

**Figure A1.** Summary of the data and methodology adopted in this study.

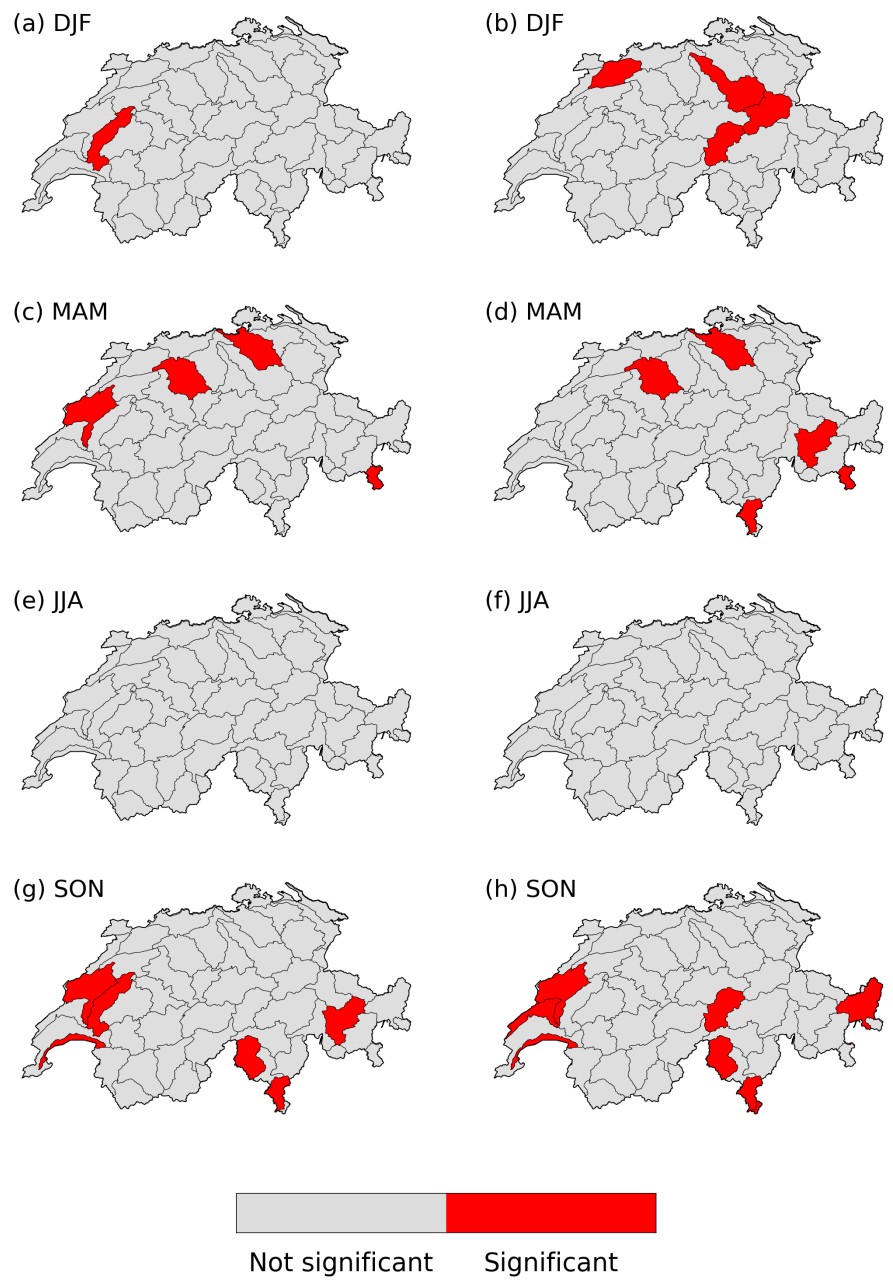

**Figure A2.** (a,c,e,g) Clustering significance in RhiresD for a time window of 5-15 days, in (a) DJF, (c) MAM, (e) JJA and (g) SON. (b,d,f,g) Same, but for a time window of 25-35 days.

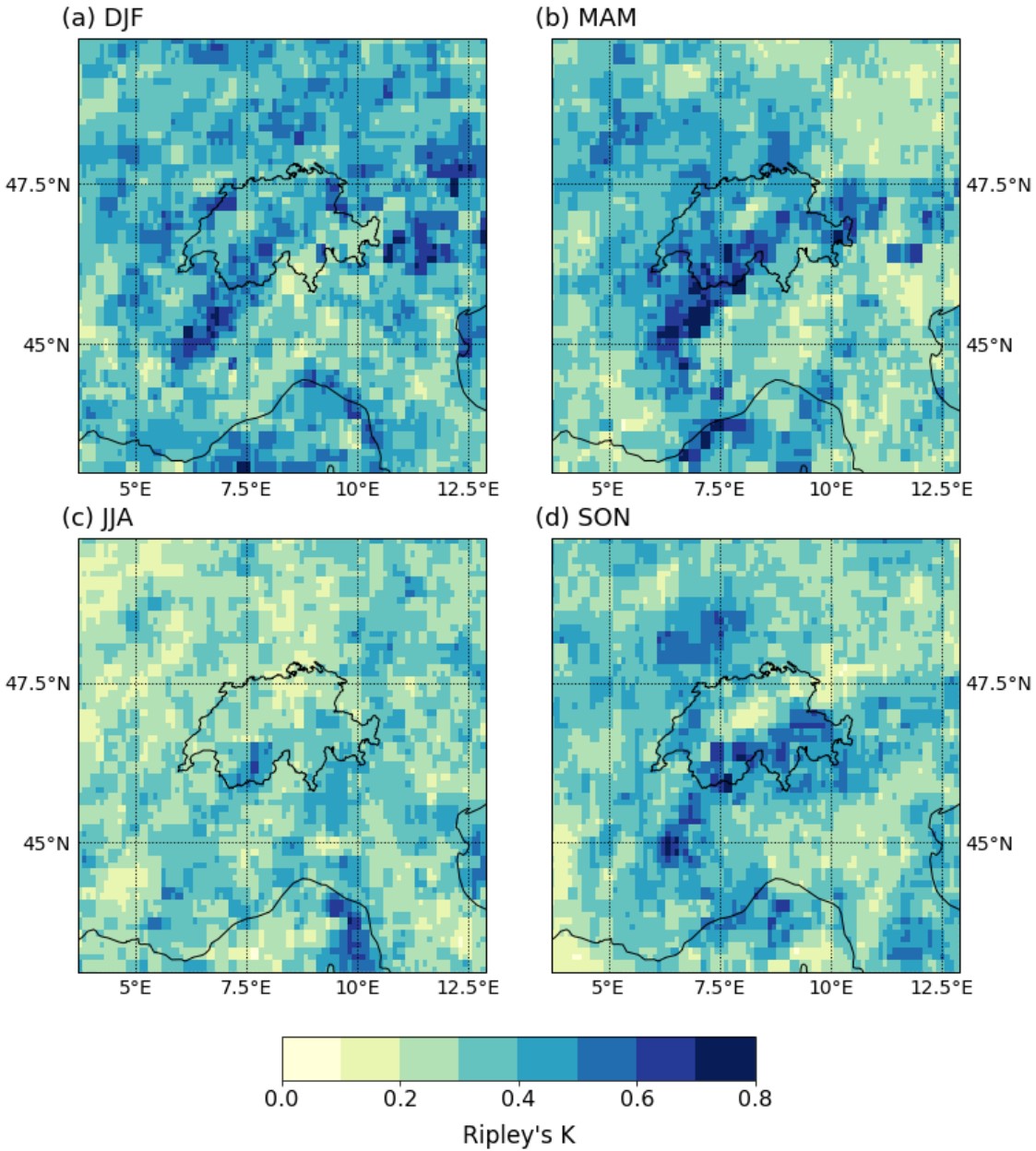

**Figure A3.** Average Ripley's K value for a 20-day window across ERA5, TRMM, CMORPH, CPC and EOBS in (a) DJF, (b) MAM, (c) JJA and (d) SON. For this comparison, all datasets were regridded to the smallest 0.1° EOBS resolution.

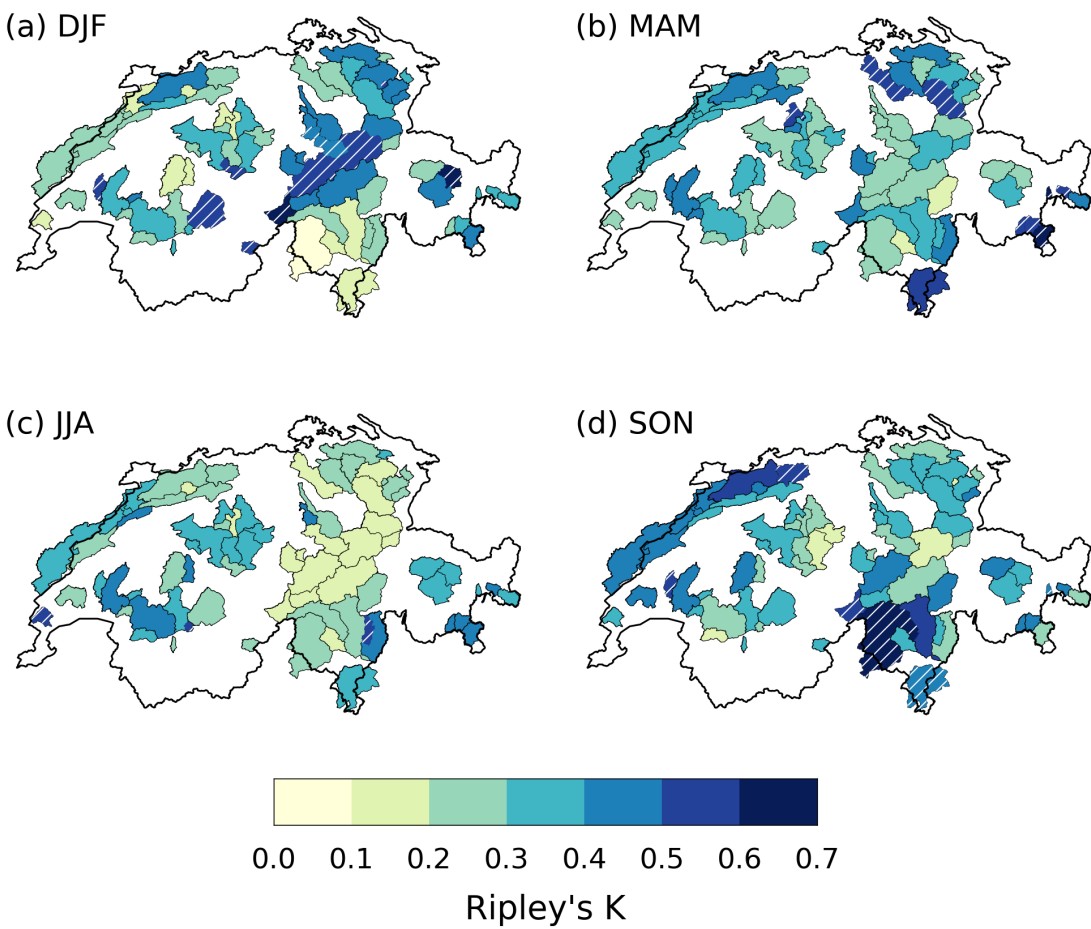

**Figure A4.** Value of Ripley's K in the RhiresD dataset for a 20-day window, in (a) DJF, (b) MAM, (c) JJA and (d) SON, for the 63-catchment partition. Compare with Figure 4.

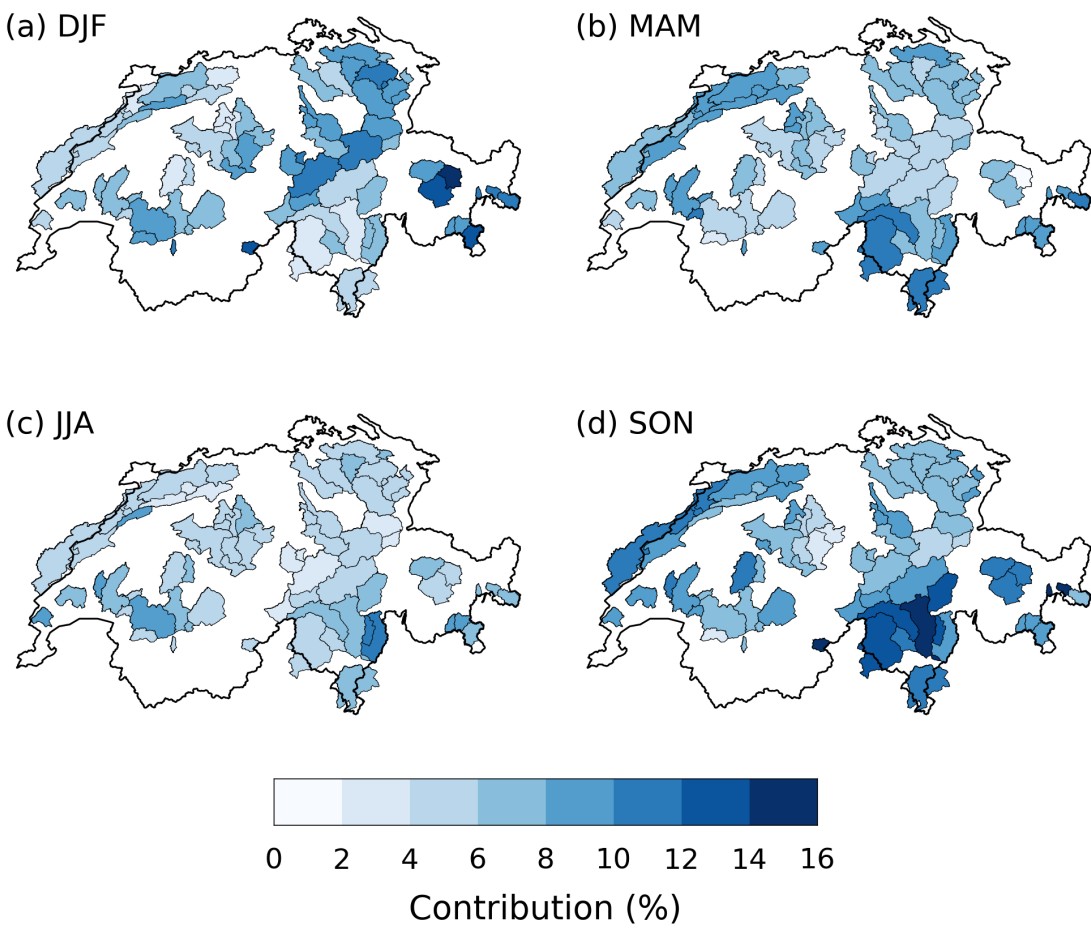

**Figure A5.** Contribution of 21-day extreme precipitation clusters to seasonal precipitation in RhiresD, in (a) DJF, (b) MAM, (c) JJA and (d) SON, for the 63-catchment partition. Compare with Figure 6.

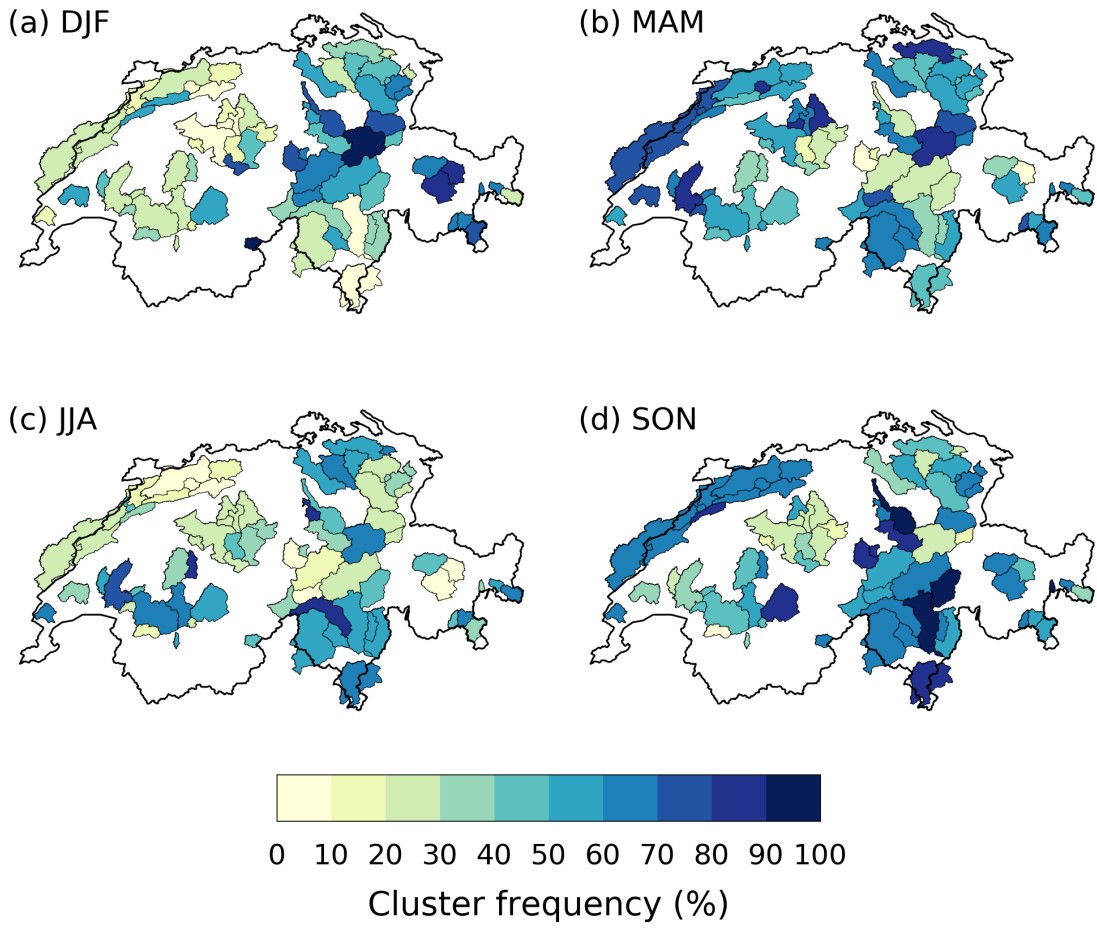

**Figure A6.** Frequency of extreme precipitation cluster occurrence during extreme 21-day cumulative precipitation events ($> 99^\text{th}$ percentile) in RhiresD, in (a) DJF, (b) MAM, (c) JJA and (d) SON, for the 63-catchment partition. Compare with Figure 7.

**Table A1.** List and main characteristics of the 93 gauged Swiss catchments used in this study.

| ID | Station | River | Area (km$^2$) | Mean elev. | Min elev. | Max elev. | Glaciation (%) | Period |
|------|---------|-------|------|------|------|------|------|--------|
| 2020 | Bellinzona | Ticino | 1517.5 | 1679 | 220 | 3345 | 0 | 1961/1–2016/3 |
| 2033 | Ilanz | Vorderrhein | 774 | 2026 | 685 | 3557 | 1.8 | 1961/1–2015/5 |
| 2034 | Payerne | Broye | 415.9 | 724 | 368 | 1574 | 0 | 1961/1–2016/12 |
| 2044 | Andelfingen | Thur | 1701.6 | 773 | 354 | 2431 | 0 | 1961/1–2016/12 |
| 2056 | Seedorf | Reuss | 833.2 | 2005 | 432 | 3598 | 6.4 | 1961/1–2016/12 |
| 2070 | Emmenmatt | Emme | 443 | 1072 | 562 | 2161 | 0 | 1961/1–2016/12 |
| 2078 | Le Prese | Poschiavino | 167.7 | 2161 | 962 | 3875 | 3.9 | 1974/1–2017/12 |
| 2084 | Ingenbohl | Muota | 316.6 | 1364 | 425 | 2731 | 0 | 1961/1–2016/12 |
| 2087 | Andermatt | Reuss | 190.2 | 2276 | 1125 | 3598 | 2.9 | 1961/1–2015/6 |
| 2104 | Weesen | Linth | 1061.5 | 1580 | 416 | 3557 | 1.6 | 1961/1–2017/12 |
| 2106 | Muenchenstein | Birs | 887.3 | 733 | 256 | 1424 | 0 | 1961/1–2016/12 |
| 2112 | Appenzell | Sitter | 74.4 | 1254 | 445 | 2431 | 0 | 1961/1–2016/12 |
| 2122 | Moutier | Birse | 185.8 | 927 | 493 | 1424 | 0 | 1961/1–2016/4 |
| 2126 | Waengi | Murg | 80.1 | 654 | 456 | 1113 | 0 | 1961/1–2016/3 |
| 2132 | Neftenbach | Toess | 343.3 | 659 | 380 | 1298 | 0 | 1975/7–2016/12 |
| 2141 | Tiefencastel | Albula | 529 | 2127 | 837 | 3317 | 0.5 | 1961/1–2016/3 |
| 2151 | Oberwil | Simme | 343.7 | 1639 | 778 | 3208 | 2.4 | 1961/1–2017/7 |
| 2155 | Wiler | Emme | 924.1 | 871 | 430 | 2161 | 0 | 1961/1–2017/9 |
| 2159 | Belp | Guerbe | 116.1 | 849 | 508 | 2128 | 0 | 1961/1–2016/12 |
| 2160 | Broc | Sarine | 636.3 | 1501 | 674 | 3207 | 0 | 1961/1–2017/12 |
| 2167 | Ponte Tresa | Tresa | 609.1 | 805 | 198 | 2207 | 0 | 1961/1–2016/3 |
| 2176 | Zuerich | Sihl | 342.6 | 1047 | 402 | 2223 | 0 | 1961/1–2015/6 |
| 2179 | Thoerishaus | Sense | 351.2 | 1076 | 524 | 2182 | 0 | 1961/1–2016/12 |
| 2181 | Halden | Thur | 1085 | 914 | 445 | 2431 | 0 | 1965/1–2016/4 |
| 2185 | Chur | Plessur | 264.4 | 1865 | 545 | 2923 | 0 | 1961/1–2015/12 |
| 2202 | Liestal | Ergolz | 261.2 | 591 | 296 | 1181 | 0 | 1961/1–2016/12 |
| 2203 | Aigle | Grande Eau | 131.6 | 1566 | 384 | 3167 | 0.8 | 1961/1–2016/3 |
| 2210 | Ocourt | Doubs | 1275.4 | 960 | 407 | 1448 | 0 | 1961/1–2016/4 |
| 2219 | Oberried | Simme | 34.7 | 2335 | 1075 | 3208 | 22.6 | 1961/1–2016/4 |
| 2232 | Adelboden | Allenbach | 28.8 | 1855 | 1093 | 2833 | 0 | 1961/1–2016/12 |
| 2256 | Pontresina | Rosegbach | 66.5 | 2701 | 1720 | 3981 | 21.7 | 1961/1–2016/3 |
| 2262 | Pontresina | Berninabach | 106.9 | 2608 | 1783 | 3981 | 14.4 | 1961/1–2016/3 |
| 2270 | Combe des Sarrasins | Doubs | 998.5 | 985 | 553 | 1448 | 0 | 1961/1–2016/3 |
| 2276 | Isenthal | Grosstalbach | 43.9 | 1810 | 767 | 2961 | 6.7 | 1961/1–2016/12 |
| 2299 | Erstfeld | Alpbach | 20.7 | 2181 | 629 | 3129 | 19.7 | 1961/1–2015/5 |

| ID | Station | River | Area (km²) | Mean elev. | Min elev. | Max elev. | Glaciation (%) | Period |
|----|---------|-------|-----------|-----------|-----------|-----------|---------------|--------|
| 2300 | Euthal | Minster | 59.1 | 1352 | 642 | 2223 | 0 | 1961/1–2016/4 |
| 2303 | Jonschwil | Thur | 492.9 | 1027 | 535 | 2431 | 0 | 1966/1–2017/10 |
| 2304 | Zernez | Ova dal Fuorn | 55.3 | 2333 | 1666 | 3114 | 0 | 1961/1–2016/4 |
| 2305 | Herisau | Glatt | 16.7 | 836 | 624 | 1145 | 0 | 1961/1–2016/4 |
| 2307 | Sonceboz | Suze | 127.2 | 1044 | 634 | 1595 | 0 | 1961/1–2017/10 |
| 2308 | Goldach | Goldach | 50.4 | 840 | 391 | 1245 | 0 | 1961/2–2016/12 |
| 2312 | Salmsach | Aach | 47.4 | 476 | 391 | 609 | 0 | 1961/4–2016/12 |
| 2319 | Zernez | Ova da Cluozza | 26.9 | 2361 | 1468 | 3115 | 0 | 1961/7–2016/4 |
| 2321 | Pregassona | Cassarate | 75.8 | 991 | 272 | 2198 | 0 | 1962/6–2016/4 |
| 2342 | Brig | Saltina | 76.5 | 2017 | 661 | 3407 | 2.5 | 1966/1–2016/12 |
| 2343 | Huttwil | Langeten | 59.9 | 765 | 566 | 1123 | 0 | 1966/1–2015/5 |
| 2355 | Davos | Landwasser | 183.7 | 2223 | 1453 | 3180 | 0 | 1967/1–2015/5 |
| 2356 | Cavergno | Riale di Calneggia | 23.9 | 1982 | 645 | 2866 | 0 | 1967/1–2016/3 |
| 2366 | La Roesa | Poschiavino | 14.1 | 2286 | 1707 | 3012 | 0 | 1970/1–2016/4 |
| 2368 | Locarno | Maggia | 926.9 | 1534 | 191 | 3208 | 0 | 1985/1–2017/12 |
| 2369 | Yvonand | Mentue | 105.3 | 683 | 436 | 946 | 0 | 1971/1–2017/7 |
| 2370 | Le Noirmont | Doubs | 1046.7 | 985 | 503 | 1448 | 0 | 1971/1–2016/3 |
| 2372 | Mollis | Linth | 600.2 | 1737 | 427 | 3557 | 2.9 | 1961/1–2015/6 |
| 2374 | Mogelsberg | Necker | 88.1 | 962 | 604 | 1513 | 0 | 1972/1–2016/4 |
| 2386 | Frauenfeld | Murg | 213.3 | 596 | 381 | 1113 | 0 | 1974/1–2016/4 |
| 2409 | Eggiwil | Emme | 124.4 | 1283 | 562 | 2161 | 0 | 1975/1–2016/4 |
| 2412 | Vuippens | Sionge | 43.4 | 872 | 674 | 1457 | 0 | 1975/2–2017/12 |
| 2415 | Rheinsfelden | Glatt | 417.4 | 506 | 340 | 1105 | 0 | 1976/1–2015/5 |
| 2419 | Reckingen | Rhone | 214.3 | 2301 | 1307 | 3598 | 11.8 | 1975/1–2015/8 |
| 2420 | Lumino | Moesa | 471.9 | 1668 | 229 | 3169 | 0 | 1980/3–2016/4 |
| 2426 | Mels | Seez | 106.1 | 1796 | 469 | 3073 | 0 | 1966/1–2017/9 |
| 2432 | Ecublens | Venoge | 227.6 | 694 | 372 | 1662 | 0 | 1979/1–2016/4 |
| 2434 | Olten | Duennern | 233.8 | 714 | 390 | 1383 | 0 | 1977/10–2015/6 |
| 2450 | Zofingen | Wigger | 366.2 | 662 | 419 | 1393 | 0 | 1979/1–2015/6 |
| 2461 | Magliaso | Magliasina | 34.4 | 927 | 269 | 1904 | 0 | 1980/1–2016/4 |
| 2468 | St. Gallen | Sitter | 261.1 | 1045 | 445 | 2431 | 0 | 1980/7–2016/4 |
| 2469 | Hondrich | Kander | 490.7 | 1846 | 558 | 3675 | 5.1 | 1980/8–2017/12 |
| 2471 | Murgenthal | Murg | 183.4 | 659 | 410 | 1123 | 0 | 1980/6–2017/12 |
| 2474 | Buseno | Calancasca | 120.5 | 1930 | 503 | 3169 | 0.2 | 1961/1–2016/4 |
| 2477 | Zug | Lorze | 100.2 | 822 | 411 | 1556 | 0 | 1982/4–2017/12 |

| ID | Station | River | Area (km$^2$) | Mean elev. | Min elev. | Max elev. | Glaciation (%) | Period |
|---|---|---|---|---|---|---|---|---|
| 2478 | Soyhieres | Birse | 569.5 | 811 | 380 | 1424 | 0 | 1982/6–2017/12 |
| 2479 | Delemont | Sorne | 213.9 | 785 | 408 | 1326 | 0 | 1982/6–2017/12 |
| 2480 | Boudry | Areuse | 377.7 | 1084 | 427 | 1573 | 0 | 1961/1–2017/6 |
| 2481 | Buochs | Engelberger Aa | 228 | 1605 | 432 | 3137 | 2.5 | 1961/1–2016/12 |
| 2486 | Vevey | Veveyse | 64.5 | 1108 | 372 | 1959 | 0 | 1984/1–2017/12 |
| 2487 | Werthenstein | Kleine Emme | 311.5 | 1171 | 525 | 2290 | 0 | 1984/4–2017/12 |
| 2491 | Buerglen | Schaechen | 107.9 | 1722 | 436 | 3221 | 1.5 | 1985/6–2017/12 |
| 2493 | Gland | Promenthouse | 119.8 | 1035 | 372 | 1667 | 0 | 1985/9–2017/12 |
| 2494 | Pollegio | Ticino | 443.8 | 1794 | 277 | 3120 | 0 | 1986/5–2017/12 |
| 2497 | Nebikon | Luthern | 104.7 | 754 | 474 | 1393 | 0 | 1988/1–2017/12 |
| 2498 | Castrisch | Glenner | 380.9 | 2014 | 685 | 3345 | 1.1 | 1988/6–2017/12 |
| 2500 | Ittigen | Worble | 67.1 | 678 | 494 | 954 | 0 | 1988/6–2017/12 |
| 2603 | Langnau | Ilfis | 187.4 | 1047 | 681 | 2045 | 0 | 1989/4–2017/12 |
| 2604 | Biberbrugg | Biber | 31.9 | 1008 | 602 | 1515 | 0 | 1989/6–2017/12 |
| 2605 | Lavertezzo | Verzasca | 185.1 | 1663 | 463 | 2837 | 0 | 1989/9–2017/12 |
| 2607 | Oberwald | Goneri | 38.4 | 2378 | 1353 | 3120 | 4 | 1990/8–2017/12 |
| 2609 | Einsiedeln | Alp | 46.7 | 1161 | 660 | 1783 | 0 | 1991/2–2017/12 |
| 2610 | Vicques | Scheulte | 72.7 | 797 | 419 | 1292 | 0 | 1992/1–2017/12 |
| 2612 | Lavertezzo | Riale di Pincascia | 44.5 | 1713 | 463 | 2520 | 0 | 1992/7–2017/12 |
| 2617 | Muestair | Rom | 128.5 | 2188 | 1167 | 3196 | 0 | 1994/5–2017/12 |
| 2629 | Agno | Vedeggio | 99.9 | 921 | 198 | 2198 | 0 | 2004/1–2017/12 |
| 2630 | Sion | Sionne | 27.6 | 1575 | 485 | 3084 | 0 | 2006/10–2017/12 |
| 2634 | Emmen | Kleine Emme | 478.3 | 1058 | 425 | 2290 | 0 | 1961/1–2015/6 |