# Peer review of "A climatology of sub-seasonal temporal clustering of extreme precipitation in Switzerland and its links to extreme discharge"

_Natural Hazards and Earth System Sciences, 2021_

## Community Comment (CC1)

**Comment on: "A climatology of sub-seasonal temporal clustering of extreme precipitation in Switzerland and its impacts"**

Dear Alexandre and Olivia,

I think that analyzing the link between temporal precipitation clustering and flood occurrence and duration is important because it helps to improve our understanding of important flood drivers. The establishment of such a link is an interdisciplinary research effort involving analyses of climatological and hydrological data. While I generally appreciate the analyses presented in this paper, I think that the consistency and link between the climatological and hydrological analyses could/should be improved by unifying methodology across variables and by better embedding the study's findings in the hydrological literature. I would like to highlight a few points, which I consider to be important from a hydrologist's point of view:

1. Threshold choice: You use a seasonally varying quantile threshold for precipitation while they use a fixed annual quantile threshold for streamflow. I think that threshold choice should be consistent and that the use of a fixed instead of a variable threshold would be more sensible for the given application as the occurrence of high-flows might be more directly related to absolute than relative exceedances. The results might substantially depend on this important methodological choice. Mixing variable and fixed thresholds does in my opinion not make sense. In any case, assessing the sensitivity of the results to the choice of threshold type (variable vs. fixed) would be highly desirable and facilitate the interpretation of your results. In addition, some of the precipitation-related results also seem to refer to exceedances of annual 99% quantiles (e.g. Fig. 2).

2. Region definition: The precipitation analysis is performed for a different set of regions than the catchments selected for the high-flow analysis (at least partially from Fig 4). In order to allow for a direct comparison of the results obtained from the two analyses (precipitation vs. discharge), it would be desirable to use the same catchment delineation used for the hydrological analysis also for the precipitation analysis. Such an analysis would be straightforward as areal precipitation sums for the 93 catchments could be derived from the gridded precipitation data set used for the analysis.

3. Catchment selection: The study is based on 93 selected catchments. It would be important to point out how and why this sub-selection was made (l. 95-101).

4. Persistent flood periods (l. 137): I would rather call these something like 'high-flow periods' as a period of 30 days is likely to contain several potentially independent events. Furthermore, L and N seem to be mixed up in the equation as L must be > N if the temporal resolution of the data is daily. If you would like to look at events, I would apply some event definition where a flood has a defined start and end.

5. Results: It would be valuable to link the results in addition to the climatological literature also to the hydrological literature about flood seasonality, flood generation processes, … E.g. l.165-167: literature on antecedent conditions and the interplay between different flood drivers; l. 282-290: literature related to rain-on-snow events; l. 288-289: literature on flood volumes and peak-volume dependencies; L. 267: literature to regime types.

6. Term flood risk: This paper only addresses the hazard part of risk and I would therefore talk about hazard rather than risk.
7. Flood recession timescales: how are they defined (l. 278)?
8. Figures: I would recommend to reconsider color choices for figures, i.e. use continuous scales for continuous variables and diverging scales only for data with a logical break point (e.g. decreases vs. increases). Furthermore, the figure captions are a bit too short and it would be helpful if you could provide more detailed descriptions of what is displayed in the figures (also what the subpanels refer to).

I hope that you find some of these comments helpful to strengthen the interdisciplinary aspect of your work.

Best regards,

Manuela Brunner

**A few potentially useful references to strengthen the hydrological aspects:**

Recently published report about floods in Switzerland: https://www.research-collection.ethz.ch/handle/20.500.11850/458556

Addor, N., Rössler, O., Köplin, N., Huss, M., Weingartner, R., & Seibert, J. (2014). Robust changes and sources of uncertainty in the projected hydrological regimes of Swiss catchments. *Water Resources Research*, *50*, 1–22. https://doi.org/10.1002/2014WR015549

Berghuijs, W. R., Harrigan, S., Molnar, P., Slater, L. J., & Kirchner, J. W. (2019). The relative importance of different flood-generating mechanisms across Europe. *Water Resources Research*, *55*, 4582– 4593. https://doi.org/10.1029/2019WR024841

Brunner, M. I., Hingray, B., Zappa, M., & Favre, A. C. (2019). Future trends in the interdependence between flood peaks and volumes: Hydro-climatological drivers and uncertainty. *Water Resources Research*, *55*, 1–15. https://doi.org/10.1029/2019WR024701

Merz, B., Nguyen, V. D., & Vorogushyn, S. (2016). Temporal clustering of floods in Germany: Do flood-rich and flood-poor periods exist? *Journal of Hydrology*, *541*, 824– 838. https://doi.org/10.1016/j.jhydrol.2016.07.041

Merz, R., & Blöschl, G. (2003). A process typology of regional floods. *Water Resources Research*, *39*(12), 1340. https://doi.org/10.1029/2002WR001952

Oppel, H., & Fischer, S. (2020). A new unsupervised learning method to assess clusters of temporal distribution of rainfall and their coherence with flood types. *Water Resources Research*, in press. https://doi.org/10.1029/2019WR026511

Wasko, C., & Nathan, R. (2019). Influence of changes in rainfall and soil moisture on trends in flooding. *Journal of Hydrology*, *575*(May), 432– 441. https://doi.org/10.1016/j.jhydrol.2019.05.054

---

## Author Comment (AC1)

**Major comments**

**Comment 1: Title**

"...and its impact" As far as I can see, you only assess the importance of clustering with regard to streamflow peaks. What are the other impacts? In the discussion you meantion "surface impact", but still do no specify.

The original title was a bit broad indeed and to make it more to the point, we suggest the following: "*A climatology of sub-seasonal temporal clustering of extreme precipitation in Switzerland and its links to extreme discharge*".

**Comment 2: Persistent floods**

For large parts of the manuscript, I was not sure what you mean by 'persistent floods'. Please consider to explain this earlier in the manuscript (introduction?) with one or two sentences. The definition only is given in section 2.2.3, but the term is used a couple of times before already. Furthermore, I am a bit confused about the definition itself. I did not fully understand it yet, I think. The 99th percentile means that 1 % of the values/days get selected. Doesn't it on an annual level mean that (depending on how you estimate the percentiles; how do you sample them by the way?) you only have 3-4 flood days per year? How can there then be 10 flood days in a 30 day period then? And how is it possible that L < N? Isn't L a time period in days and N the number of flood days in this time period? Please clarify.

The manuscript was indeed confusing on the issue of "persistent floods" and the wording has to be modified. Instead of "persistent floods" it is better to speak of "persistent high-discharge periods". To speak of "events" may be deceitful since we do not identify specific events with a beginning and end; our metric for persistent high discharge includes long-lived extreme discharge events but also recurrent, independent events. We also removed any reference to persistent high-discharge events before introducing their definition in the methods section.

It is true that the $99^{th}$ percentile of daily discharge is exceeded on average only ~3 times per year. However, the occurrence of extreme discharge events is not homogeneously distributed in time. Discharge series indeed exhibit strong temporal autocorrelation. In addition, clusters of precipitation extremes (as we show in this work) likely cause repeated exceedances of extreme discharge thresholds over short periods of time.

L and N were also in the wrong order and we corrected the mistake (L > N).

Another question in this regard: When does a year start and end in your case? Do you consider the hydrological year for Switzerland? Consider to change the section title in 2.2.3 to 'Persistent flood events'.

We do not need to define the beginning and end of a year in our analysis. Discharge percentiles are calculated using the entire available series and remain the same for all time steps.

Is there a specific reason why you choose the 99th and the 95th percentile? You refer to these events as 'floods'. However, runoff above the 95th percentile on an annual basis does not necessarily cause flooding. On 5 % of the days this value (~18 days every year) is crossed, right? Please consider to pick up this point in the discussion section.

These two percentiles are used to define extreme discharge events, but of course other values could be used too. The 95th percentile may be a bit low to speak of "extreme" discharge, but its advantage is that it allows to select more events compared to the 99th percentile. Still, the comparison of results between the two percentiles generally shows good agreement (Figures 9, 11-13).
You are also correct to point out that flooding does not necessarily occur when extreme discharge thresholds are crossed. The choice of the word "flood" in the initial manuscript version was confusing and we decided to change it to "extreme discharge".

**Comment 3: Catchments**

I am somehow missing a better overview on the catchments investigates. Something like a table (or overview graph?) summarizing information on gauge locations, names, catchment areas, discharge data availability,... Please provide more information on how you selected this set of catchments? Why does it fit to your type of analysis? Why do you need catchments with glacial, nival and pluvial regimes? Also I do not see what time frame was considered for the different watersheds. It is a bit confusing that in Fig.1 DEM and catchments have both black lines as boundaries. Also some catchments reach outside the DEM/Switzerland. There the DEM does not cover. River gauges usually are marked with reversed triangles, I think. Maybe you can color the catchment areas according to their mean elevation?

We will add the following supplementary table containing catchment characteristics:

**Table A1**. *List and main characteristics of the 93 gauged Swiss catchments used in this study.*

[revised manuscript text omitted]

Catchment selection was performed by Muelchi et al. (2021) based on several criteria: data availability, the absence of major lakes, minimal human influence and satisfactory calibration results in their hydrological model. The fact that these catchments are somewhat well

distributed across Switzerland and cover the range of climates and hydrological regimes that is typical of this country is an advantage, since it allows us to explore the potential role of extreme precipitation temporal clustering across regions, climates and hydrological regimes, and to see the limits of our analysis.

Figure 1 can also be updated to show the elevation beyond the Switzerland border, and to better show catchment boundaries and gauge locations (see Figure S1 below).

[Figure]

Figure S1 – Topography of Switzerland (shading, with major lakes shown in light blue) and gauged catchments used in this study (catchment boundary: blue lines; catchment gauge location: red triangles). The thick black line indicates the Swiss border.

**Comment 4: Comparison precipitation data sets**

The comparison of different gridded data sets is an interesting point of your study and should be already mentioned in the objectives at the end of the introduction. I think it could be interesting to see maps of Ripley's K value (Fig. 6 with K on grid level) of the different data sets in their original resolution. Please consider to try such a figure.

The manuscript includes quite a lot of figures already, but we could add such a figure to the appendix (see Figure S2 below).

[Figure]

Figure S2 – Average Ripley's K value for a 20-day window across ERA5, TRMM, CMORPH, CPC and EOBS in (a) DJF, (b) MAM, (c) JJA and (d) SON. For this comparison, all datasets were regridded to the smallest 0.1° EOBS resolution.

**Comment 5: Scheme for methods**

In my opinion, a scheme depicting your approach on how to detect temporal clusters in precipitation/discharge would help the reader to understand the methods faster and easier.

We suggest expanding the cluster period identification method to make it clearer: "*We identify extreme precipitation cluster events over 21-day time windows with the algorithm of Kopp et al. 2021 (see their Figures 3 and 4). Starting from the declustered binary extreme event series, the first step is to calculate the 21-day moving sum of extreme event counts. In a second step, we select the 21-day period with the largest event count, if that sum is larger than 2. Otherwise, no clusters are found and the algorithm stops. In the case of multiple 21-day periods with the same extreme event count, the one with the largest precipitation total is selected first. In the third step, we remove from the binary event series the extreme events that occur in the selected 21-day period. The algorithm is then run again from the first step onwards to identify the next cluster event. This procedure avoids any overlap between cluster events. The choice of the 21-day time window is well-suited to quantify clustering at sub-seasonal timescales, and is generally consistent with the length of observed cluster episodes that led to major floods in Switzerland (see introduction). Results do not differ significantly*

*for slightly shorter or longer (2-4 weeks) windows (see also Kopp et al. 2021).*
*We then characterize clusters of precipitation extremes with two metrics related to their potential impact. The first is the average contribution of cluster periods to seasonal precipitation. This contribution increases with the frequency and total precipitation of cluster periods. The second metric is the frequency of cluster periods during extreme 21-day precipitation accumulations. It gives an idea of how often cluster periods are responsible for extreme precipitation accumulations, a frequent trigger of flood events (Froidevaux et al. 2015).*"

Please explain in more detail how you calculate percentiles on a monthly basis for precipitation. Do you take only days with rainfall into account or all days (also the days with no rain)?

Regarding precipitation percentiles and the definition of extreme precipitation events, we suggest the following revision at the beginning of the methods section: "*For each dataset, precipitation extremes are defined on a monthly basis as days when daily accumulated precipitation exceeds the 99th percentile of the corresponding month. For instance, January precipitation values are compared to the January 99th percentile. The percentiles are calculated using all days (both with and without precipitation).*"

I do not really get the selection of timescales presented Page 4 Line 123. Please consider to extend the explanation on the timescale 5-15 etc.

The selection of timescales for clustering significance is motivated by our focus on sub-seasonal timescales. Please see the revised section 2.3.2:

*Temporal clustering of precipitation extremes is quantified with Ripley's K function (Ripley, 1981). We give here a quick overview of the methodology and refer the reader to Tuel and Martius (2021) for further details. For a given window size w, Ripley's K function applied to a time series measures the average number of extreme events in a neighbourhood of w days before and after a random extreme event in the series. This gives information about the tendency towards temporal clustering in the series. The larger the value of Ripley's K function for a given w, the more clustered the extreme events. The significance of temporal clustering in the series is then assessed by comparing Ripley's K values to those obtained from a Monte-Carlo sample of 5000 simulated homogeneous Poisson processes with the same average event density as the observed series. In homogeneous Poisson processes, events occur independently from each other and therefore exhibit complete temporal randomness. Because we chose monthly percentiles to define extreme precipitation events, the occurrence rate of extremes is constant throughout the year. Thus we can test for clustering significance against homogeneous series. Non-homogeneous (and more complex) series would have been required if the likelihood of extreme event occurrence has been a function of time.*
*From this comparison we get an empirical p-value for each w. As we deal with multiple hypothesis tests, we implement a false discovery rate procedure (Wilks, 2016) with a baseline*

*significance level of 5% to identify catchments where clustering is significant. Clustering significance is assessed for two intervals of w values, characteristic of sub-seasonal timescales: 15-25 and 25-35 days. Clustering is said to be significant for a given interval if it is significant for at least half of the w values in that interval.*

Also consider to compare your precipitation clusters to regular precipitation accumulation periods presented in Froideveaux et al 2015: https://doi.org/10.5194/hess-19-3903-2015. What is the difference? Is the performance in explaining flooding better?

Froidevaux et al. (2015) did not exactly look at the same kind of events as we do (they looked at annual discharge maxima, whereas we look at all exceedances of given discharge percentiles). They also did not consider the issue of flood duration. Still, they found that precipitation accumulations in the 1-2 days before annual peak discharge events were most critical in explaining these events. This is consistent with our results showing a strong discharge response peaking one day after extreme precipitation events, whether clustered on non-clustered (Figure 13). We have added the following sentence to the discussion: "*The discharge response to both clustered and non-clustered extreme precipitation events typically peaks one day after the event (Figure Fig_13), consistent with the findings of Froidevaux et al. (2015).*"

**Comment 6: Structure**

There is a bit of a mix between results and discussion, I think (see specific comments). The structure of the discussion could follow the result section. Would make it easier to read, I think.

We considerably modified the manuscript structure in response to reviews and hope that the new version will be easier to follow. In particular, one issue with the original version of the manuscript is the discussion of the seasonality in extreme precipitation and discharge magnitudes which did not belong to the results. Instead, this discussion mostly brings together elements from previous studies and we prefer to move it to the data/methods section when introducing the area of study. That way, the results and discussion sections can focus on the clustering and links to extreme discharge only.

In the discussion, you focus a lot on physical interpretation (4.1). However, I am not sure whether your study provides enough new information that allow to support/challenge any of those hypotheses. There is no need to remove it, but maybe focus more on the discussion of your actual analysis.

It is difficult to discuss the patterns of significance in spatial clustering without questioning their physical interpretation. Our study does provide new information regarding the spatio-temporal patterns of clustering significance in Switzerland, so it is worthwhile to spend some time discussing potential mechanisms. Still, we agree that the reference to the NAO and

atmospheric rivers goes beyond the scope of our discussion and we removed the corresponding sentences.

In line 291 and following you discuss flood risk. You do not address exposure and vulnerability at all. You only provide information on the hazard component, I think. Please specify what new insights you study provides with regard to flood hazard. Fig. 13: Please provide information on this in the method section, present in results and discuss later.

We replaced "risk" with "hazard" in the revision to be more consistent with our analysis. Regarding the methods, we suggest expanding the methods section with the following subsection that explains how we assess the influence of temporal clustering on discharge extremes:

"***Effects of temporal clustering of extreme precipitation on the occurrence and duration of extreme discharge***

*We analyse the influence of clusters of precipitation extremes on discharge in two ways: by looking at discharge characteristics after clusters of precipitation extremes, and at precipitation characteristics before periods of persistent high discharge.*

*First, we calculate for each catchment the average number of extreme discharge days during and up to 5 days after 21-day clusters of precipitation extremes. This number is then divided by the average number of extreme discharge days expected for periods of the same length and same time of the year as the selected cluster periods. This yields an "odds ratio" of extreme discharge occurrence after clusters of extremes. From the identification of 21-day cluster periods, precipitation extremes can also be separated into "clustered" and "non-clustered" events. We then look at the likelihood of extreme discharge occurrence after both types of events to highlight potential differences in the discharge response.*

*Second, for each of the persistent high-discharge periods identified as described previously, we calculate the number of precipitation extremes and the percentile of total accumulated precipitation in the 10 preceding days.*"

**Specific comments**

Page 1 Line 9: "...magnitudes decrease more slowly after clustered events" Compared to what other type of events?

This is compared to non-clustered events. We may rephrase as "*In addition, discharge magnitudes decrease more slowly after clustered precipitation extremes than non-clustered ones.*"

Page 3 Line 70: "[...] scale of ≈ 1000 km2 catchments covering the whole of Switzerland." How many catchments? Please consider to rephrase this sentence.

The reference to the catchments is unnecessary here and we may rephrase as "*We then discuss the patterns and robustness of the spatio-temporal distribution of sub-seasonal clustering in Switzerland.*"

Page 3 Line 72-73: This description of the outline is not necessary in my view. You follow the typical structure and section titles are clear.

You are right, we will remove the corresponding sentences.

Page 3 Line 81: "63 catchments" It is 93, right?

The original manuscript was not clear about our use of two different sets of catchments, one for the hydrological analyses (gauged catchments for which discharge data is available, but which do not cover the whole of Switzerland) and one for the statistical clustering analyses (catchments that are not necessarily gauged but which provide a full partition of Switzerland). We added a subsection to the data section to highlight this point:

"*We average RhiresD data over a hydrological partitioning of Switzerland that consists of 63 catchments with a mean area of 900 km2 (see Figure 4). Catchment-scale aggregation is useful to identify the occurrence of high-impact heavy precipitation events, and also to smooth RhiresD data to a lower resolution more consistent with its effective resolution. Though we could also use the set of 93 gauged catchments, this set does not cover the whole of Switzerland. Consequently, we opt for a countrywide partitioning of 63 larger catchments (for which no discharge observations are available). To be comprehensive and to help with the comparison of results, we also show in appendix the results obtained for the 93-catchment set.*"

Page 4 Line 104: remove "as"

Corrected, thanks

Page 5 Line 140: Please consider to add the subsection 3.1 Seasonality of heavy precipitation and floods a the catchment scale

We re-organized these paragraphs and moved them to the data/methods section under the heading "Study region". These are not results, but a discussion of already existing knowledge on the seasonality of extreme precipitation and discharge across Switzerland (illustrated by Figures 2 and 3).

Figure 2: What are panels a-d? Please clarify in figure caption.

Thanks for noticing – we had forgotten to specify that panels corresponded to the four seasons: (a) DJF, (b) MAM, (c) JJA and (d) SON.

Page 5 Line 145: "Extreme precipitation events" 60 % of precipitation events means that 60 % of the values above the 99th percentile are located in a season, right? Please consider to explain again what your definition of "extreme precipitation event" is.

We agree that the wording was confusing. We now discuss Figures 2 and 3 before the methods section that introduces the definition of extreme precipitation events so as to avoid any confusion. In these figures we look at the seasonal frequency of exceedance of the fixed 99$^{th}$ daily precipitation percentile. So, a value of 60% indeed means that 60% of exceedances on average occur within the corresponding season.

Page 5 Line 148-154: This is discussion already, I think. Please consider moving this information into the discussion section.

This whole section is in fact essentially literature review which we illustrate with figures 2 and 3. We moved it outside of the results section to avoid confusion.

Page 6 Line 2: [...] due to heavier?

The sentence was reformulated as "*Alpine catchments, especially at high elevations, are mainly driven by snow- and glacier melt*".

Page 6 Line 159: "combination of saturated of frozen soil" Where can I see this in your analysis? Please focus on the presentation of your results here and discuss later.

As with one of your previous comments, this refers to previous literature and we moved the whole paragraph outside of the results section.

Page 6 Line 160: "floods" Values above the 99th percentile are not floods. Isn't high runoff values better?

You are correct that extreme discharge and floods (in the hydrological sense of the term) are not the same. We replaced the word "floods" by extreme discharge throughout the paper to avoid any confusion.

Page 6 Line 162: Where is the "Ticino area"?

The canton of Ticino is located in southern Switzerland. Since we did not specify it on a map, we replaced all references to that region by the slightly broader "Southern Alps" which we now define on Figure 1-b.

Page 6 Line 165-168: Please consider to move this paragraph into the discussion section. Focus on your results here

As with one of your previous comments, this refers to previous literature and we moved the whole paragraph outside of the results section.

Fig 4 and 5: You have a lot of figures. Isn't it possible to combine those two figures? E.g., significant areas full color and not significant one stripes?

Good suggestion, we combined Figures 4 and 5.

Page 6 Line 183-184: Please consider to move this into the discussion section. Focus on the presentation of your results here.

Good point.

Paragraph 3.2 – I find it hard to read this paragraph. The metrics used should be described in the method section already. Why do you select those metrics? You jump a lot between figures here, I think. Please try to organize the result sections better, so it is easier to follow.

We agree and added details of the selected cluster metrics to the methods section as follows: "*We characterize clusters of precipitation extremes with two metrics related to their potential impacts. The first is the average contribution of cluster periods to seasonal precipitation. This contribution increases with the frequency and total precipitation of cluster periods. The second metric is the frequency of cluster periods in extreme 21-day precipitation accumulations. It gives an idea of how often cluster periods are responsible for extreme precipitation accumulations, a frequent trigger of flood events (Froidevaux et al. 2015)*".

Regarding paragraph 3.2, we suggest reformulating it as follows: "*We now expand the statistical analysis by showing the characteristics of clustered precipitation events. In winter, extreme precipitation clusters contribute an average of ≈10% to total winter precipitation along the Alpine ridge where clustering is statistically significant (Figure 7-a). Additionally, clusters occur during about 60-70% of extreme 21-day precipitation accumulations (above the corresponding 99th percentile) (Figure 8-a). Elsewhere, clusters contribute little both to seasonal and extreme precipitation accumulations. In spring, the average contribution of clusters to seasonal precipitation is overall weak (<10%), even for catchments where clustering in RhiresD is statistically significant (Figure 7-b). Yet, over Western Switzerland, periods of extreme 21-day accumulations are almost always cluster periods as well (Figure 8-b). In summer, consistent with the absence of clustering at that time of the year, clusters are not contributing much to seasonal precipitation. Finally, in fall, cluster contribution to seasonal precipitation reaches its annual maxima of 12-16% over Southeastern Switzerland (particularly the Southern Alps). It is also quite high (≥10%) over Western Switzerland where clustering is statistically significant as well (Figure 7-d). In addition, more than 80% of extreme precipitation accumulation periods are accompanied by cluster events in the Southern Alps (Figure 8-d). Since extreme discharge in this area are most common during fall (Figure 3-d), this suggests a possibly important role of extreme precipitation clusters in high-impact weather events in this region and at that time of the year.*"

Page 7 Line 207: "Unsurprisingly"?

The word could be deleted.

Page 7 Line 209: Start new sentence here.

A new sentence could indeed be added: "*This is a clear signature of the glacial/nival runoff regime dominance above that altitude.*"

Page 7 Line 214-216: Please concentrate on the description of your results here. Discuss later.

These sentences still describe results (Figure 13) but should be reformulated to make it clearer. Please see revised section 3.3 "Discharge response to extreme precipitation clustering".

Page 7 Line 219: "20-30 %". This seems low to me. What about the rest?

This figure is not low – note that we are not considering probabilities that sum to 1 here (there is no "rest" to consider). Instead, we look at exceedances of the $99^{th}$ discharge percentile during the 5 days following an extreme event. So, 20-30% of days means that on average the $99^{th}$ discharge percentile will be exceeded during 1-1.5 days after an event. $99^{th}$ percentiles are exceeded on average ~3 times per year, so our results imply an important effect. We added a sentence to point this out in the text.

Page 7 Line 220: "The occurence [...]" Move to discussion?

This sentence is still describing results (Figure 12), but is discussed further in the discussion.

Page 8 Line 221-225: This is a key sentence of your study, I think.

Yes, and we now include it in the abstract as well.

Page 8 Line 240: What is "IVT"?

Sorry for that oversight. IVT stands for 'Integrated Vapour Transport'. We propose to reformulate the sentence as follows: "*During winter, extreme precipitation events in northern Switzerland usually occur in connection with extreme integrated water vapour transport with convergence onto the orography, for instance linked to atmospheric rivers.*"

Page 8 Line 249: What is "PV"?

PV (potential vorticity) was defined at l. 153 but since we use the acronym only twice we can simply say "potential vorticity".

Section 4.2 title: Replace "flood risk" with "flooding"?

We can replace "flood risk" by "flooding hazard".

Page 9 Line 264: Where do you show the analysis on "major floods"?

We do not show it explicitly; with this sentence we argue that our results suggest a role for temporal clustering in major flood events. Still, in the original manuscript the sentence was misplaced. We suggest moving it to the paragraph of section 4.2 discussing the case of southern Switzerland.

Page 9 Line 266: "runoff regimes at lower elevation" You mean pluvial-type rivers? Rainfall-dominated? You say rainfall is important in rainfall-dominated rivers?

We did mean pluvial-type regimes and we will specify it in the revision.

Page 9 Line 284: needs to be focus of further research?

Good suggestion.

Page 10 Line 300: Why is the same water level in winter more damaging than in summer?

We realise that this sentence was confusing. We did not intend to say that the same water level is necessarily more damaging in winter than in summer. Rather, in this sentence we argue that to define floods based on fixed discharge percentiles (no seasonal variation) potentially overlooks some high-discharge events outside of summer in glaciated/snow-driven catchments. So it isn't that the same water level is necessarily more damaging in winter than in summer, but that very high discharge events in winter (just below the summer peaks) may still be quite damaging. We suggest reformulating the text as follows: "*While from the perspective of impacts it makes sense to define floods based on annual discharge percentiles, in snow-driven or glaciated catchments, this choice may discard potential high-discharge conditions occurring outside summer.*"

Page 10 Line 307: "Clustering is most significant over the Alps in winter." Why?

Good question! It is beyond the scope of this study, which only seeks to characterize the statistics of temporal clustering, but obviously it raises the question of why clustering occurs more frequently over the Alps in winter and the Southern Alps in fall. Most likely the clustering is related to the persistence of specific regional/North Atlantic weather patterns leading to intense moisture transport towards the Alps.

Figure 9: How/Why do you calculate the average cumulative precipitation quantile?

We had not explained this calculation in detail. To analyse the influence of clusters of precipitation extremes on discharge, we take two approaches: the first is to look at discharge characteristics after clusters of precipitation extremes, and the second to look at precipitation before periods of persistent high discharge. Cumulative precipitation percentiles are useful to characterise the precipitation before periods of persistent high discharge. In practice, for each of the persistent high-discharge periods (identified using (L,N) values), we calculate the number of precipitation extremes and the percentile of total accumulated precipitation in the

10 preceding days. We added these details to the methods section under "Effects of temporal clustering of extreme precipitation on the occurrence and duration of extreme discharge".

Figure 10: Describe panel b) in figure caption.

Sorry for the oversight. Panel (b) is the same as panel (a) but for the number of extreme precipitation events.

---

## Author Comment (AC2)

**Major comments**

We thank the reviewer for their helpful comments that helped to improve the clarity of the paper.

**Comment 1: Title**

I didn't get how the cluster events are identified (section 2.2.2). The authors cite Kopp et al 2021, which I looked at, but actually I still don't fully get it. Anyway this is an important variable of the study and I think the article should be self-sufficient.

We suggest rephrasing this important paragraph as follows: "*We identify extreme precipitation cluster events over 21-day time windows with the algorithm of Kopp et al. (2021) (see their Figures 3 and 4). Starting from the declustered binary extreme event series, the first step is to calculate the 21-day moving sum of extreme event counts. In a second step, we select the 21-day period with the largest event count (i.e., the highest number of extreme events), if that count is larger than 2. Otherwise, no clusters are found and the algorithm stops. In the case of multiple 21-day periods with the same extreme event count, the one with the largest precipitation total is selected first. In the third step, we remove from the binary event series the extreme events that occur in the selected 21-day period. The algorithm is then run again from the first step onwards to identify the next cluster event. This procedure avoids any overlap between cluster events. The choice of the 21-day time window is well-suited to quantify clustering at sub-seasonal timescales, and is generally consistent with the length of observed cluster episodes that led to major floods in Switzerland (see introduction). Results do not differ significantly for slightly shorter or longer (2-4 weeks) windows (see also Kopp et al. 2021).*"

**Comment 2**

I got confused with the analysis of flood days: how is it possible to get 5 days exceeding the 99th quantile within 10 days (Figure 9)? This is very unlikely: there are on average 3.65 exceedances per year.

Daily discharge series exhibit strong autocorrelation, particularly in the extremes. This is due to the catchment response time being longer than that of precipitation. Exceedances of the 99th percentile are therefore not randomly distributed, and sequences of multiple exceedances over short time periods are not uncommon.

**Comment 3**

Finally, there are many figures for a quite short article. Some figures are little commented (e.g., those of section 3.3) and perhaps they could be omitted to make it more concise (not mandatory).

We suggest merging Figures 4 and 5.

**Specific comments**

title : « impacts » is confusing because it relates to social sciences

We propose "*A climatology of sub-seasonal temporal clustering of extreme precipitation in Switzerland and its links to extreme discharge*".

l 38  and others: Tuel and Martius 2021 is in review so I coulnd't check

The paper was just published in Weather and Climate Extremes and is available at https://www.sciencedirect.com/science/article/pii/S2212094721000426.

l 111 : please consider explaining the declustering procedure (and its goal) in short

We suggest expanding this sentence as follows: "*As the individual weather systems associated with extreme precipitation may sometimes last for several days, we remove the short-term temporal dependence in the occurrence of extreme precipitation events by applying a standard runs declustering procedure (Coles 2001) with a run length of 2 days, well-suited for Switzerland (Barton et al. 2016). The goal of the declustering is to remove short-term dependence and to identify independent events. This procedure is for example applied prior to a peak-over-threshold statistical analysis. The declustering merges extreme events that are separated by less than 2 days into a single event.*"

l 124 : « at least half of the n values » : is it the same « n » as the window size above ? (I don't think so). Do you consider all time scales between e.g. 5-15 days ? (i.e. 5, 6, 7, … , 15 days)

It is not the same "n" and to avoid confusion we suggest replacing the variable by "w". As you say, for each time interval, we do look at all values in the interval. To make that clearer we suggest the following revision: "Clustering significance is assessed for two intervals of *w* values, characteristic of sub-seasonal timescales: 15-25 and 25-35 days. Clustering is said to be significant for a given interval if it is significant for at least half of the *w* values in that interval."

all of section 2.2.2 : unclear to me even with Kopp et al. Please clarify.

We suggest the following revision to make this part easier to follow: "*We identify extreme precipitation cluster events over 21-day time windows with the algorithm of Kopp et al. (2021) (see their Figures 3 and 4). Starting from the declustered binary extreme event series, the first step is to calculate the 21-day moving sum of extreme event counts. In a second step, we select the 21-day period with the largest event count (i.e., the highest number of extreme events), if that count is larger than 2. Otherwise, no clusters are found and the algorithm stops. In the case of multiple 21-day periods with the same extreme event count, the one with the largest precipitation total is selected first. In the third step, we remove from the binary*

*event series the extreme events that occur in the selected 21-day period. The algorithm is then run again from the first step onwards to identify the next cluster event. This procedure avoids any overlap between cluster events. The choice of the 21-day time window is well-suited to quantify clustering at sub-seasonal timescales, and is generally consistent with the length of observed cluster episodes that led to major floods in Switzerland (see introduction). Results do not differ significantly for slightly shorter or longer (2-4 weeks) windows (see also Kopp et al. 2021).*

*We then characterize clusters of precipitation extremes with two metrics related to their potential impact. The first is the average contribution of cluster periods to seasonal precipitation. This contribution increases with the frequency and total precipitation of cluster periods. The second metric is the frequency of cluster periods during extreme 21-day precipitation accumulations. It gives an idea of how often cluster periods are responsible for extreme precipitation accumulations, a frequent trigger of flood events in Switzerland (Froidevaux et al. 2015)*"

section 2.2.3 « flood days » may be confusing → heavy discharge days ?

Indeed, the word "floods" was confusing and we replaced it by "extreme discharge" throughout the paper.

l 138 : I guess (L,N) should be (N,L)

The order was indeed reversed – now corrected!

l 141 : please add a subsection here
l 148 « This seasonality… end of paragraph → please consider moving it into the discusion section

We moved this paragraph away from the results section since it was essentially a discussion of already existing knowledge, illustrated by Figures 2 and 3.

l 160 : « floods are rare » : it's actually hard to tell because the color scales are different in Figs 2 and 3. Please consider merging these two figures and using the same color range.

We now use the same color range for the two figures (0-100%).

Section 3.2: I missed it because I didn't get the definition of clusters

We hope the proposed revision for section 2.2.2 is now clear enough.

l 227 : « not very different between clustered and non-clustered extremes » : actually the y-scales are different and we can read quite different values for the two cases (about 0.7 vs 0.4). Please consider using the same y-scale.

The comparison between clustered and non-clustered extremes should be done on each panel separately (the two panels correspond to different daily discharge thresholds: 95th and 99th percentiles).

l 267 : « runoff regime » : please clarify

We refer to pluvial regimes here, which we will specify in the revision.

l 61 : brackets

Corrected.

l 81 : 63 → 93

There was confusion about the catchment number in the original manuscript. We suggest making it clearer that we are using two distinct sets of catchments: a hydrological partitioning of the whole of Switzerland that includes 63 (not necessarily gauged) catchments, and one with 93 gauged catchments which we use in the discharge analyses. The reason why we use these two sets is that the second does not cover the whole country. For completeness and to make it easier to compare results of the clustering and discharge analyses, we will add figures to the appendix showing the clustering results for the 93-catchment set. We added a subsection to the data section to highlight this point:
"*We average RhiresD data over a hydrological partitioning of Switzerland that consists of 63 catchments with a mean area of 900 km² (see Figure 4). Catchment-scale aggregation is useful to identify the occurrence of high-impact heavy precipitation events, and also to smooth RhiresD data to a lower resolution more consistent with its effective resolution. Though we could also use the set of 93 gauged catchments, this set does not cover the whole of Switzerland. Consequently, we opt for a countrywide partitioning of 63 larger catchments (for which no discharge observations are available). To be comprehensive and to help with the comparison of results, we also show in appendix the results obtained for the 93-catchment set.*"

l 82 : smooth

Corrected.

l 104 : as as

Corrected.

l 105 « all-day percentiles » : confusing to me all-day // monthly

'all-day' meant that the percentiles were calculated from all the days in the corresponding month and not just from the wet days. We can reformulate as follows: "*For each dataset,*

*precipitation extremes are defined on a monthly basis as days when daily accumulated precipitation exceeds its 99th percentile of the corresponding month. For instance, January precipitation values are measured against the January 99th percentile. The percentiles are calculated using all days (both with and without precipitation).*"

l 160 « floods » → please specify « in Jura »

Good point.

l 240 : IVT acronym

Sorry for that oversight. IVT stands for 'Integrated Vapour Transport'. We propose to reformulate the sentence as follows: "*During winter, extreme precipitation events in northern Switzerland usually occur in connection with extreme integrated water vapour transport with convergence onto the orography, for instance linked to atmospheric rivers.*"

l 263  flood risk → hazard

Good point.

Fig 1 : please locate Ticino, Jura, … (also all the Swiss maps are elongated)

All figures were updated with a Mercator projection. We also added a second panel to Figure 1 showing Switzerland's main geographical regions referred to in the main text: Jura, Plateau, Alps and Southern Alps (and we now avoid referring to the "Ticino region"; instead we talk about the Southern Alps).

Fig 2 , 3: please specify the seasons a,b,c,d

Sorry for the oversight, the captions should include (a) DJF, (b) MAM, (c) JJA and (d) SON.

---

## Author Comment (AC3)

We thank Dr. Brunner for the valuable comments that have significantly improved the clarity of the paper and highlighted important points to take up in the discussion.

I think that analyzing the link between temporal precipitation clustering and flood occurrence and duration is important because it helps to improve our understanding of important flood drivers. The establishment of such a link is an interdisciplinary research effort involving analyses of climatological and hydrological data. While I generally appreciate the analyses presented in this paper, I think that the consistency and link between the climatological and hydrological analyses could/should be improved by unifying methodology across variables and by better embedding the study's findings in the hydrological literature. I would like to highlight a few points, which I consider to be important from a hydrologist's point of view:

**Comment 1**
Threshold choice: You use a seasonally varying quantile threshold for precipitation while they use a fixed annual quantile threshold for streamflow. I think that threshold choice should be consistent and that the use of a fixed instead of a variable threshold would be more sensible for the given application as the occurrence of high-flows might be more directly related to absolute than relative exceedances. The results might substantially depend on this important methodological choice. Mixing variable and fixed thresholds does in my opinion not make sense. In any case, assessing the sensitivity of the results to the choice of threshold type (variable vs. fixed) would be highly desirable and facilitate the interpretation of your results. In addition, some of the precipitation-related results also seem to refer to exceedances of annual 99% quantiles (e.g. Fig. 2).

Our choice of seasonally-varying percentiles to define extreme precipitation can indeed be confusing, all the more so as extreme discharge events are selected using fixed percentiles. We did not sufficiently justify this choice in the original manuscript version. It is motivated by two reasons.
First, the statistical significance of the clustering is more difficult to assess when the extreme events series is non-homogeneous, i.e., when the likelihood of extreme events has a seasonal cycle. In theory, it would be possible to estimate that cycle and use it to generate non-homogeneous Poisson series to test for clustering significance. However, it introduces more uncertainty and seasonal variations in clustering significance would be overlooked. As we see from our results these are substantial.
Second, seasonal variations in extreme precipitation/discharge occurrence are not necessarily aligned, and in fact they are not over much of Switzerland. In the Jura, extreme discharge occurs preferentially during winter when the magnitude of precipitation extremes is lower than in summer and fall. Similarly, the larger precipitation extremes over the Swiss Plateau in summer are not accompanied by significantly more frequent extreme discharge. Choosing fixed percentiles to define precipitation extremes is therefore not ideal. Our methodology is admittedly constrained by the fact that we take a country-wide approach and try to analyse regions with different climates and hydrological regimes.
We suggest reformulating the methods section relative to the definition of extremes by adding the following: "*Three reasons justify the choice of seasonally-varying thresholds for*

*precipitation and fixed thresholds for discharge. First, such a choice removes the influence of the seasonality in extreme precipitation magnitude. The occurrence rate of extreme precipitation events is therefore constant across the year, and detecting clustering significance is straightforward. Second, impacts of discharge extremes are usually related to their absolute rather than relative magnitude. Third, the seasonal cycles of extreme precipitation and discharge magnitudes are not in phase over much of Switzerland (Figures 2 and 3). The most extreme discharge does not necessarily occur after the heaviest precipitation events. Surface conditions, like soil saturation, presence of snow/ice, vegetation cover, or evaporative demand, considerably shape the discharge response to heavy precipitation (Paschalis et al. 2014). As they vary substantially from one season to the next, the discharge response to the same precipitation magnitude may differ depending on the season.*"

**Comment 2**

Region definition: The precipitation analysis is performed for a different set of regions than the catchments selected for the high-flow analysis (at least partially from Fig 4). In order to allow for a direct comparison of the results obtained from the two analyses (precipitation vs. discharge), it would be desirable to use the same catchment delineation used for the hydrological analysis also for the precipitation analysis. Such an analysis would be straightforward as areal precipitation sums for the 93 catchments could be derived from the gridded precipitation data set used for the analysis.
Catchment selection: The study is based on 93 selected catchments. It would be important to point out how and why this sub-selection was made (l. 95-101).

We use two sets of catchments, one for the statistical precipitation analyses, the other for the extreme discharge analyses, for the reason that the set of 93 gauged catchments does not cover the whole country. This is why we use a hydrological partitioning of Switzerland with 63 catchments. However, we agree that this may make it difficult to compare results from both sets of analyses. Hence, we will include in the appendix the results for the 93-catchment set corresponding to Figures 4, 5, 7 and 8. We will also make it more explicit why we choose these two sets of catchments.
Regarding catchment selection: this point was raised by other reviewers as well. Catchment selection was performed by Muelchi et al. (2021) based on several criteria: data availability, the absence of major lakes, minimal human influence and satisfactory calibration results in their hydrological model. They are well distributed across Switzerland and cover the range of climates and hydrological regimes that is typical of this country. This is an advantage since it allows us to explore the potential role of extreme precipitation temporal clustering across regions, climates and hydrological regimes, and to see the limits of our analysis. We will include this information in the revised version, along with an appendix table containing catchment details (river, area, elevation, etc.)

**Comment 3**

Persistent flood periods (l. 137): I would rather call these something like 'high-flow periods' as a period of 30 days is likely to contain several potentially independent events.

Furthermore, L and N seem to be mixed up in the equation as L must be > N if the temporal resolution of the data is daily. If you would like to look at events, I would apply some event definition where a flood has a defined start and end.

The manuscript was indeed confusing on the issue of "persistent floods" and the wording has to be modified. Your suggestion of "persistent high-flow periods" is good and we propose to adopt it in the revised version. To speak of "events" may be deceitful since we do not identify specific events with a beginning and end. Our metric for persistent high flow can thus include long-lived extreme discharge events but also recurrent, independent events. L and N were also in the wrong order and we corrected the mistake (L > N); thank you for pointing it out.

**Comment 4**
Results: It would be valuable to link the results in addition to the climatological literature also to the hydrological literature about flood seasonality, flood generation processes, ... E.g. l.165-167: literature on antecedent conditions and the interplay between different flood drivers; l. 282-290: literature related to rain-on-snow events; l. 288-289: literature on flood volumes and peak-volume dependencies; L. 267: literature to regime types.

Thank you for this comment. In the initial manuscript version, the discussion of our results was not clearly separated from that of the seasonality in extreme discharge and precipitation, which could lead to some confusion. We suggest separating the two: first, a review of the literature on the seasonality of extreme discharge and precipitation in Switzerland in a "Study region" section; and second, a discussion of the links between our results and the hydrological literature in the Discussion section (see below).

"*Switzerland can be divided into several regions with distinct climates and hydrological regimes: the Jura, the Plateau, the Alps and the Southern Alps (Figure 1-b) (MeteoSwiss 2013, Aschwanden and Weingartner 1985). These regions notably exhibit quite different seasonal cycles in extreme precipitation and discharge occurrence. In the Plateau, the heaviest precipitation occurs chiefly during summer (Figure 2-c) (Helbling et al. 2006, Diezig et al. 2007, Panziera et al. 2018), as a result of convective instability (Stucki et al. 2012), frequent westerly winds and Atlantic water vapour transport (Giannakaki et al. 2016). In summer, however, evapotranspiration is highest and soils are less saturated than in the cold season. Consequently, extreme discharge events are about equally likely to occur in winter, spring and summer (Figure 3). In the Jura, while the magnitude of extreme precipitation events still peaks in summer, its seasonality is less pronounced. About 20% of extreme precipitation events indeed occur in winter and spring each (Figure 2), triggered by forced orographic ascent of moist westerlies (Froidevaux and Martius 2016). Extreme discharge, however, is mostly confined to winter and spring, largely driven by rain-on-snow processes (Diezig et al. 2007, Helbling et al. 2006, Koplin et al. 2014).*
*As in the Jura, the seasonal cycle in extreme precipitation occurrence over the Alps is not strong (Figure 2) (Frei and Schär 1998, MeteoSwiss 2013}. The peak is reached in summer and fall for most catchments, when extreme precipitation occurs as a result of local convective instability (Stucki et al. 2012), but winter and spring still concentrate 30-40% of*

*extreme events. The outlook for discharge is very different, however. Alpine catchments, especially at high elevations, are mainly driven by snow- and glacier melt (Aschwanden and Weingartner 1985). Thus, extreme discharge is almost exclusively confined to summer (Figure 3-c) (Koplin et al. 2014, Muelchi et al. 2021b). Finally, the Southern Alps experience extreme precipitation mostly during summer and fall (Figure 2-c,d) (Frei and Schär 1998, Isotta et al. 2014). Such behaviour results from the frequent southerly advection of moist Mediterranean air caused by upper-level troughs (Barton et al. 2016). These atmospheric conditions are connected to potential vorticity streamers or cut-offs centred west of the Alps, which are most frequent during fall (Martius et al. 2006). Extreme discharge in this region also occurs primarily during fall (50-60% of events; Figure 3-d)."*

[Figure]

**Figure 1.** (a) Topography of Switzerland (shading, with major lakes shown in light blue) and gauged catchments used in this study (catchment boundary: blue lines; catchment gauge location: red triangles). The thick black line indicates the Swiss border. (b) Switzerland's topography (shaded) and major climate/hydrological regions (red).

Regarding links of our results to the hydrological literature, we can expand the discussion as follows:

*"Still, from the perspective of surface impacts, clusters remain relevant, regardless of their overall frequency, if they increase flood hazard. The discharge response to both clustered and non-clustered extreme precipitation events typically peaks one day after the event (Figure 12), consistent with the findings of Froidevaux et al. (2015). However, our results show that clusters of precipitation extremes strongly impact the likelihood of occurrence and the duration of high-discharge events, particularly at low elevations (Figures 10 and 11). This influence is noticeably larger than for non-clustered precipitation extremes (Figure 12). On average, daily accumulated precipitation during clustered and non-clustered extremes is similar. Instantaneous precipitation rates might be different, but it is not possible to verify it given the daily resolution of the precipitation data. However, the first extreme in a cluster event likely increases soil moisture, which enhances the discharge response to the subsequent precipitation extremes (Merz et al., 2006; Nied et al., 2014; Paschalis et al., 2014). The role of antecedent soil moisture on flood generation and volume is well-documented for*

*Switzerland and Alpine catchments (e.g., Keller et al., 2018). This may explain why extreme discharge probability decreases more slowly after clustered precipitation extremes compared to non-clustered events (Figure 12).*

*This difference is quite high in the Southern Alps (e.g., Figure 9-c,d), possibly due to the fact that floods in this area generally occur in the fall (Figure 3-d; Barton et al. (2016)) when clusters bring substantial amounts of precipitation (Figure 7-d). There, frequent clusters leading to extreme precipitation accumulations are likely to be an important precursor of major flood events, as confirmed by observations of several damaging clustering periods (Barton et al., 2016). This region of Switzerland also experiences the largest precipitation extremes (Umbricht A, 2013). Additionally, it is characterised by poor infiltration rates, steep slopes and weak soils (Aschwanden and Weingartner, 1985). Infiltration excess (connected to Hortonian-type storm runoff generation) may therefore be more rapidly reached than in the rest of the country. Coupled with saturation excesses following the first extreme event in a cluster, it might explain why the region stands out in most of our analyses. By contrast, in the Alps during winter, though clustering is statistically significant, its impact on extreme discharge is quite limited. This results most likely from the fact that discharge in Alpine catchments is lowest in winter, when much of the precipitation falls as snow and the magnitude of precipitation extremes is generally lower.*

*Finally, the case of Western Switzerland during spring is interesting. Though rare, clusters are responsible for almost all extreme precipitation accumulations (Figure 7-b). Over this region, floods are somewhat less frequent in spring than in winter (Figure 3-a,b), despite similar extreme precipitation likelihood (Figure 2-a,b). This may result from fewer rain-on-snow events, a major flood process for the region (Aschwanden and Weingartner, 1985; Köplin et al., 2014) but also drier soils coupled to high infiltration rates (Aschwanden and Weingartner, 1985). Yet, spring floods can still be quite devastating, since precipitation generally falls as rain instead of snow, and limited vegetation cover makes erosion more likely. Consequently, cluster events that affect Western Switzerland during spring should be the focus of further research.*"

We know little about the literature on flood volume and peak-volume dependence, and would be grateful if you could point us to relevant papers.

**Comment 5**
Term flood risk: This paper only addresses the hazard part of risk and I would therefore talk about hazard rather than risk.

We agree and would refer to hazard only in the revised version.

**Comment 6**
Flood recession timescales: how are they defined (l. 278)?

"Recession timescale" has a very specific meaning in hydrology which we do not use here. Instead, it would be preferable to rephrase the sentence as "*This may explain why the*

*likelihood of extreme discharge occurrence decreases noticeably slower after clustered extremes than after non-clustered extremes.*"

**Comment 7**
Figures: I would recommend to reconsider color choices for figures, i.e. use continuous scales
for continuous variables and diverging scales only for data with a logical break point (e.g. decreases vs. increases). Furthermore, the figure captions are a bit too short and it would be helpful if you could provide more detailed descriptions of what is displayed in the figures (also what the subpanels refer to).

Thank you for this comment. We will update most of the figures to avoid diverging colour bars for continuous variables.

---

## Author Response (AR2)

**Reviewer #1 comments**

**Major comments**

**Comment 1: Structure**
Please consider to move 'Study area' to the beginning of Chapter 2. Maybe make a new chapter called 'Study area and data' and a new separate chapter 3 'Methods'.

The difficulty is that the "Study area" section includes information from the datasets presented under "Data" (Figures 2 and 3). It would make sense, however, to divide into two chapters as suggested: "Data and study area" and "Methods".

**Comment 2: Map Ripley's K**
I am not sure why you do not (also) make the analysis on the grid-cell level of RhiresD. Such a map could be quite informative and provide more detailed spatial information. Such a figure would be better resolved as maps on sub-basin level.

RhiresD is given on a 2x2km grid scale, but its effective resolution is on the order of 15x15km (15km being the typical inter-station distance). Therefore, an analysis at the 2x2km grid scale might give a false sense of confidence. In addition, the clustering is more relevant for flood hazard at the catchment scale (100-1000km$^2$) rather than the grid-point scale (1-10km$^2$).

**Comment 3: Figure header**
I realized that it was a bit annoying to have to search in the figure captions what seasons the individual figure panels represent. Is it possible to add a header such as a) DJF? Or does this go against journal figure guidelines?

We agree and it doesn't seem to be in contradiction with figure guidelines. All corresponding figures were updated accordingly.

**Comment 4: Scheme to explain methods**
You use a lot of data and quite some different analytical tools. It was not easy to keep an overview on the individual numbers, windows, variables, watersheds, regions,... you use/calculate. Maybe consider making a scheme that illustrates your workflow. It would really help, I think. I do not think that such a scheme it is a requirement, but it would facilitate the readability of the manuscript.

Please see the updated Figure A1. As we already have many figures, we added it to the supplemental material.

**Specific comments**

L68-75 Maybe also say here that you also use different rainfall products.

Good suggestion, we expanded the second sentence: "*First, we aim to quantify the sub-seasonal clustering of precipitation extremes in time across Switzerland using several gridded station- and satellite-based datasets.*"

L34: flood hazard

Corrected, thanks.

L218: I do not really see this 'significant temporal clustering of precipitation extremes is generally found along the Alpine ridge' in Fig 4a or A1 b. Only a fraction of the Alps (as defined in Fig. 1b) shows significant signals. You also write this in the conclusion (L365). Please consider to be more specific here.

The clustering over the Alps in winter is less obvious in RhiresD than it is in the other gridded datasets, which indicate a wide region of clustering significance stretching along the Alpine ridge from southwestern to eastern Switzerland (Figure 5-a). Still, you are right to point out that the signal is not as clear in RhiresD. Thus, we suggest reformulating the sentence you refer to as follows: "*In winter, significant temporal clustering of precipitation extremes is mainly found in central Switzerland, along the Alpine ridge, at the 15-25 and 25-35 day timescales*". We also suggest reformulating the sentence comparing clustering in winter with the other precipitation datasets as follows: "*Clustering significance over the Alps in winter is also present in the coarser-resolution data, but with a wider extent than in RhiresD.*" For the conclusion, we suggest the following statement: "*Various station- and satellite-based datasets point to generally significant clustering over the Alps in winter, particularly their central part, and over Southern Switzerland during fall.*"

L134 and L184: Better do not refer to your result figures in the method section already. This was a bit confusing for me.

At line 184 we refer to the figures of Kopp et al. (2021), not to ours. At line 160 we refer to figures 2 and 3 which are not part of our results.

L212: Why 10 days? Can you add one sentence here why you choose 10 days? You use a lot of different windows (5, 10, 21,…), so it is important to give a bit more justification, I think.

Froidevaux et al. (2015) showed that while flood timing was mainly associated with extreme precipitation accumulations in the 2-3 days before the flood event, many floods were nonetheless associated with wet conditions (not "extreme wet") in the 4-14 days before the event. These longer-lasting wet conditions could possibly play a role in the persistence of high discharge, hence why we choose to start calculations 10 days before the beginning of

persistent high-discharge periods. We added the following sentence to the text: "*We choose to begin 10 days before because Froidevaux et al. (2015) showed that moderate wet conditions occurred in the week preceding many flood events in Switzerland, hence the need to look beyond the few days preceding persistent high-discharge periods.*"

L223: You actually do not compare the different rainfall products, right?

We limit our comparison of the different datasets to superposing their clustering significance maps and identifying areas of agreement.

L224: closing bracket missing

Corrected, thanks.

Fig A2: K displayed here is calculated as the average of of all K form the different data sets? Why? It would be interesting the see a comparison between the rainfall products? Does this make sense? Did you try this? Is it out of scope. Please justify.

The point of figure A2 is to give a sense of the average number of expected extreme precipitation events in the neighbourhood of any random extreme event (hence why we show the average Ripley's K value). But one shouldn't draw too much from it. Because the different datasets have different lengths, the same Ripley's K value will not be associated with the same level of significance in each dataset. Additionally, we limit our comparison of the various datasets to identifying regions where they agree on the presence/absence of temporal clustering significance. We do not seek to explain why a given region is significant in one dataset and not in another. This would go a bit far for the present study. Each dataset also has its own biases and could tend to over- or underestimate clustering significance in different regions. Looking at significance across the datasets therefore allows to highlight regions where they tend to agree, regardless of potential biases and data sources.

L225: Fig. 5 b is not white. Hence at least one, sometimes two or three datasets, suggest significant clustering. Please consider to rephrase this sentences as 'there a no signs' seems a bit strong.

You are correct, we need to qualify this statement. We suggest to reformulate as follows: "*Temporal clustering during spring is generally less significant across Switzerland (Figure 6-b). Two datasets indicate significant clustering locally in northwestern Switzerland, but none do along the northern and southern borders where significance was found in RhiresD.*"

Fig. 9 caption: It should be 'within a 10-day (respectively 20-day, 30-day) window […]'

Thanks, corrected.

Fig. 11: Can you explain a bit more what is in this figure. There only is a short description in the method section, but I think it is an important figure and would deserve another one or two sentences of explanation and description.

The figure shows the odds ratio of extreme discharge occurrence in the period during and up to 5 days after cluster events. The equation for the odds ratio is given section 3.4, but to make it clearer we suggest expanding this section by adding the following: "*The odds ratio compares the likelihood of extreme discharge occurrence in the presence of a precipitation cluster to its likelihood in the absence of a precipitation cluster. The higher it is, the stronger the relationship between the occurrence of extreme discharge and precipitation clusters.*"

L279: SST is sea surface temperature? Please also add a reference here.

You are right, we removed the acronym. Tuel and Martius (2021) discuss the relevance of sea-surface temperatures for temporal clustering at various timescales.

L314: Change 'strongly impact' to 'increase'?

Good suggestion, thanks.

L335: As you investigate on a seasonal level, lower runoff (compared to other seasons) is no explanation for low impact of precipitation clusters on discharge, I think. The main reason is snowfall, as you also mention. Rainfall just is solid and stored in snowpacks.

Precipitation extremes are investigated on a seasonal level, but not discharge extremes which are defined based on fixed percentiles. Thus, discharge extremes are by construction much less frequent in winter than in summer in snow- and glacier-dominated catchments.

L351: 'heavily biased': As far as I can see, you only investigate the hazard component. As you never had the ambition to assess risk in the first place, I would try not spend too much time in the discussion on this.

Yes, the point of this short paragraph is to underline that we only look at the hazard component of risk, not exposure or vulnerability. We may shorten it as follows: "*Our results only focus on the hazard component of flood risk. We do not take into account exposure and vulnerability, which may differ substantially between catchments due to variability in population density, infrastructure, flood management capacities, etc.*"

Conclusion: Maybe you can add a sentence on the other rainfall products used? Are they also suitable for such analysis...

The third sentence can be expanded as follows: "*Various station- and satellite-based datasets point to generally significant clustering over the Alps in winter, particularly their central part, and over Southern Switzerland during fall.*"

**Reviewer #2 comments**

**Minor comments**

l. 79 : do you have a reference for RhiresD dataset ?

The main reference for RhiresD is Frei and Schär (1998) (interpolation algorithm) which we cite already. We suggest expanding the last sentence of the first paragraph in the "Precipitation" section as follows: "A detailed description of this dataset can be found at https://www.meteoswiss.admin.ch/home/climate/swiss-climate-in-detail/raeumliche-klimaanalysen.html and the interpolation algorithm is described in Frei and Schär (1998)."

l. 84 : please specify the resolution of ERA5

The spatial resolution of ERA5 is 0.25°.

l. 96 : please specify the year of Muelchi et al.

We updated the reference with the year (2021).